# GABAergic motor neurons bias locomotor decision-making in *C. elegans*

Ping Liu [1,2], Bojun Chen[1] & Zhao-Wen Wang [1✉]

Proper threat-reward decision-making is critical to animal survival. Emerging evidence indicates that the motor system may participate in decision-making but the neural circuit and molecular bases for these functions are little known. We found in *C. elegans* that GABAergic motor neurons (D-MNs) bias toward the reward behavior in threat-reward decision-making by retrogradely inhibiting a pair of premotor command interneurons, AVA, that control cholinergic motor neurons in the avoidance neural circuit. This function of D-MNs is mediated by a specific ionotropic GABA receptor (UNC-49) in AVA, and depends on electrical coupling between the two AVA interneurons. Our results suggest that AVA are hub neurons where sensory inputs from threat and reward sensory modalities and motor information from D-MNs are integrated. This study demonstrates at single-neuron resolution how motor neurons may help shape threat-reward choice behaviors through interacting with other neurons.

---

[1] Department of Neuroscience, University of Connecticut School of Medicine, Farmington, CT 06030, USA. [2] Department of Pathophysiology, School of Basic Medicine, Tongji Medical College, Huazhong University of Science and Technology, Wuhan, China. ✉email: zwwang@uchc.edu

Perceptual decision-making is a cognitive process of choosing one action from various alternatives based on sensory information. The ability to produce appropriate choice behaviors in face of conflicting environmental factors, such as rewards and threats, is crucial to animal survival. The neural substrates for perceptual decisions may be broadly divided into three components: sensory system, decision-making system, and motor control system. Increasing evidence suggests that the motor system is not a mere passive recipient of commands from the decision-making system, but can interact with the sensory and decision-making systems to shape the final decision. For example, in humans and monkeys, the motor system is an integral component in oculomotor decision-making[1–6]. In mice, locomotion modulates neuronal activities in the visual and auditory cortices[7,8]. In flies, activities of visual neurons are regulated by walking and flying behaviors[9,10]. In worms, head undulation regulates steering movements toward an olfactory attractant through sensorimotor integration[11,12]. Although a role of the motor system in decision-making is increasingly appreciated, little is known about the neural circuit and synaptic bases of this function.

The nematode *C. elegans* is a powerful model for investigating the circuit and gene bases of behavior because its complex behaviors are produced by a nervous system of only 302 neurons, for which a complete map of neural connections has been produced. In their natural environments and under laboratory conditions, worms navigate toward food and attractive chemical odors but avoid toxic chemicals and hyperosmotic conditions. The neural circuits mediating many sensory responses have been described. For example, the attractive odor diacetyl is detected by a pair of sensory neurons known as AWA[13,14] whereas a noxious hyperosmotic condition caused by either glycerol or fructose is detected by a pair of sensory neurons known as ASH[15,16]. The AWA and ASH sensory neurons then activate downstream neural circuits mediating forward and backward movements, respectively[17–19]. Major components of the forward neural circuit include the bilateral pair of AVB premotor interneurons and B-type cholinergic motor neurons (B-MNs) whereas those of the backward circuit include the bilateral pair of AVA premotor interneurons and A-type cholinergic motor neurons (A-MNs) (Fig. 1a)[20–24]. AVA play crucial roles in producing variable sensory responses and in sensorimotor integration[25,26]. Although ultrastructural data indicate the presence of chemical synapses from GABAergic motor neurons (D-MNs) to AVA[27], physiological significance of this synaptic connection is unknown.

In this study, we investigated synaptic interactions of D-MNs with threat and reward neural circuits, and with premotor interneurons and upstream cholinergic motor neurons using a multifaceted approach. We demonstrate a physiological role of D-MNs in threat-reward decision-making with evidence at the circuit, behavioral, cellular, and molecular levels. At the circuit and behavioral levels, we found that D-MNs tilt the balance of threat-reward decision-making toward reward behavior by inhibiting AVA interneurons, and that electrical synapses between the left and right AVA interneurons help reach this biased decision by balancing inhibitory synaptic signals from D-MNs. At the cellular level, we found that D-MNs are activated by both excitatory inputs from cholinergic MNs and current flow through a stretch receptor in their cell membrane, and that the left and right AVA interneurons differ significantly in electrical properties. At the molecular level, we identified key postsynaptic receptors mediating synaptic transmission from cholinergic MNs to D-MNs and from D-MNs to AVA interneurons, a stretch-activated mechanoreceptor in D-MNs, and an innexin responsible for the electrical coupling between the two AVA interneurons. Collectively, our results depict a circuit and molecular picture showing how a biased threat-reward decision may be reached through sensory integration in AVA interneurons and retrograde regulation of AVA by D-MNs.

## Results

**Cholinergic MNs control GABAergic MNs through LGC-46 receptor.** To understand how D-MNs contribute to neural circuit function and behavior, it is important to know how they are controlled by upstream neurons, which are cholinergic MNs (Fig. 1a). An earlier study reported that the postsynaptic receptor in D-MNs is ACR-12[28]. To confirm the role of ACR-12, we determined whether spontaneous postsynaptic currents (PSCs) in D-MNs depend on cholinergic MNs and ACR-12. In VD5, a representative of D-MNs innervating ventral body-wall muscles, spontaneous PSCs occurred at a rate of $5.9 \pm 1.3$/s and had a mean amplitude of $6.0 \pm 0.6$ pA in wild type, and knockdown of *unc-17* (vesicular acetylcholine (ACh) transporter) specifically in cholinergic neurons caused 65% decrease in spontaneous PSC frequency with the remaining events having a smaller mean amplitude (Fig. 1b). Given that there might be residual *unc-17* expression in the knockdown strain, these results suggest that the majority, if not all, of the spontaneous PSCs in VD5 are due to synaptic transmission from cholinergic MNs. To assess the role of ACR-12 in the synaptic transmission from cholinergic MNs, we compared spontaneous PSCs in both VD5 and DD3 (a representative of D-MNs innervating dorsal body-wall muscles) between wild type and *acr-12(ok367)*, a putative null resulting from a deletion[29]. We obtained this mutant from the Caenorhabditis Genetics Center, confirmed its molecular lesion by PCR, and outcrossed it with wild type three times before the electrophysiological experiments. Unexpectedly, in both VD5 and DD3, the frequency and mean amplitude of spontaneous PSCs were similar between wild type and the mutant (Supplementary Fig. 1). We also found that the amplitude of exogenous ACh-induced whole-cell current in VD5 was similar between wild type and the mutant (Supplementary Fig. 2). These results suggest that ACR-12 is not a significant postsynaptic ACh receptor in D-MNs, at least under our experimental conditions.

We used VD5 as the representative of D-MNs in subsequent experiments for simplicity. To identify a candidate for the postsynaptic ACh receptor, we tested the effect of exogenous ACh on VD5 whole-cell current in mutants of genes encoding either nicotinic ACh receptor-like subunits or cys-loop ligand-gated ion channel subunits[30,31]. Besides the *acr-12* mutant described above, mutants of 12 other genes, including *acr-2, acr-9, acr-14, acr-18, acr-20, lgc-26, lgc-40, lgc-46, acr-16, unc-29, unc-38,* and *unc-63,* were chosen for the analysis based mostly on reported expression in motor neurons. ACh-induced current was normal in mutants of all these genes (Supplementary Fig. 2) except for *lgc-46*. In *lgc-46(ok2900)* and *lgc-46(ok2949)*, which are deletion mutants involving one or two exons (www.wormbase.org), ACh-induced whole-cell current was decreased by >70% compared with wild type (Fig. 1c). To determine whether LGC-46 is a postsynaptic receptor in D-MNs, we compared spontaneous PSCs in VD5 between wild-type and the *lgc-46* mutants. The mutants showed ~95% decrease in spontaneous PSC frequency with the rare remaining events having a smaller mean amplitude (Fig. 1b). These mutant phenotypes could be rescued by expressing wild-type LGC-46 specifically in GABAergic neurons, and produced in wild-type worms by GABAergic neuron-targeted *lgc-46* RNAi (Fig. 1b, c). GABAergic neuron targeting was achieved through the use of an *unc-47* (vesicular GABA transporter) promoter, which has activities in 26 neurons, including RME (4), AVL (1), RIS (1), DVB (1), and all the D-MNs (13 VDs and 6 DDs)[32]. In addition, we examined the effect

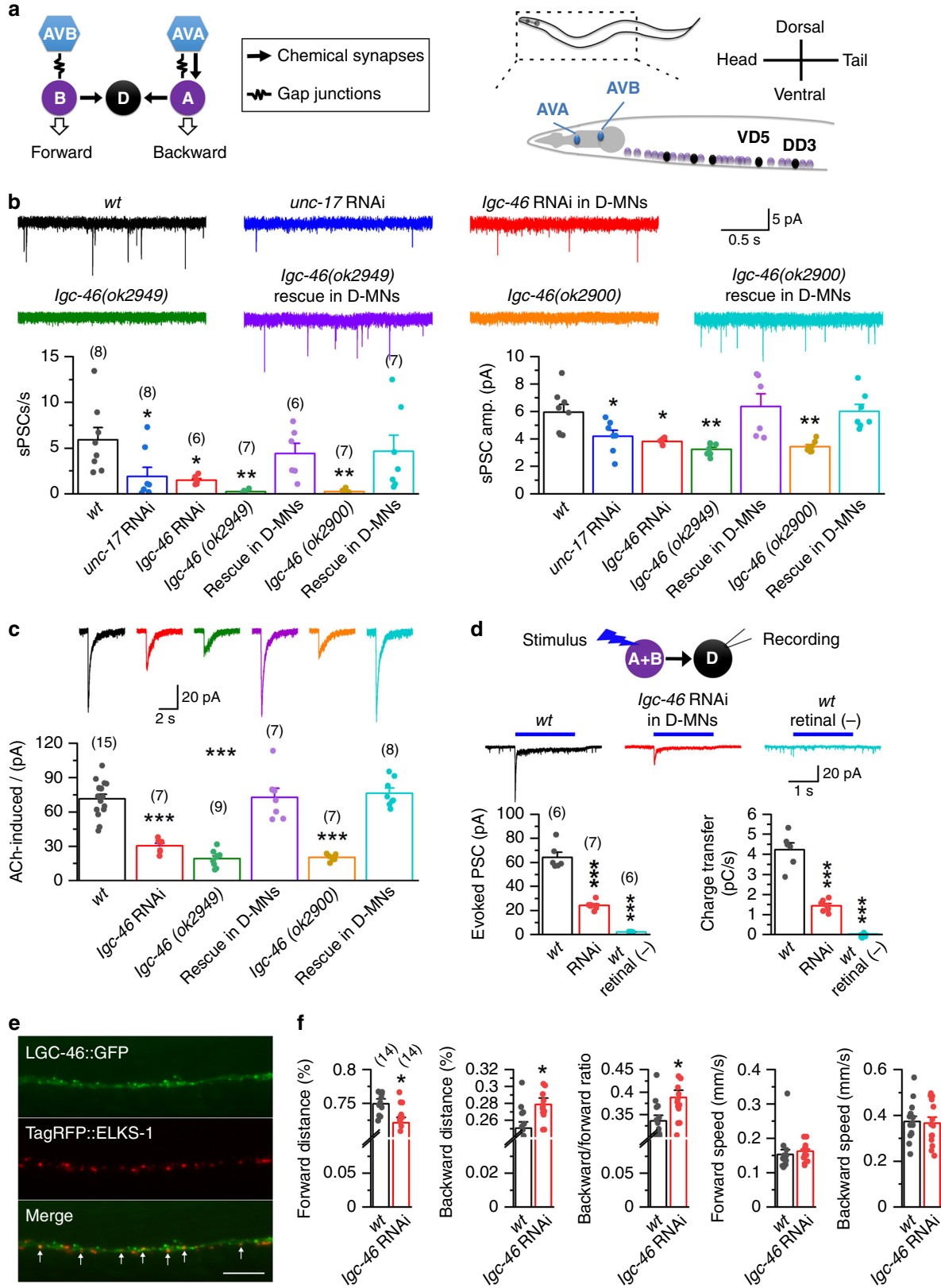

of optogenetic activation of cholinergic neurons on evoked PSCs in VD5 with worms expressing channelrhodopsin-2 (ChR2) under the control of *unc-17* (vesicular ACh transporter). Blue light stimulus evoked a large initial transient current followed by a smaller sustained current in wild type but had a much smaller (~65% less) effect in the *lgc-46* RNAi strain (Fig. 1d). We also

analyzed LGC-46 subcellular localization in D-MNs by expressing GFP-tagged LGC-46 in worms in which presynaptic sites of cholinergic MNs were labeled by TagRFP::ELKS-1[33]. In the transgenic worms, GFP showed both diffuse and punctate localization in neurites of D-MNs, with some of the GFP puncta being either colocalized with or juxtaposed to the presynaptic

**Fig. 1 LGC-46 is a key component of a postsynaptic acetylcholine (ACh) receptor in D-MNs. a** Diagram depicting major components of the forward and backward circuits, and their synaptic connections. A- and B-type cholinergic motor neurons are labeled A and B, respectively, whereas GABAergic motor neurons are labeled D. AVA and AVB are premotor command interneurons for backward and forward movements, respectively. **b** Spontaneous postsynaptic currents (sPSCs) in VD5 (held at −60 mV) depended on LGC-46 and ACh release. *unc-17* and *lgc-46* were knocked down in cholinergic neurons using P*unc-17* and GABAergic neurons using P*unc-47*, respectively. **c** Whole-cell current caused by exogenous ACh (1 mM) in VD5 (held at −60 mV) depended on LGC-46. Compared with wild type (*wt*), $p = 0.000$ *lgc-46* RNAi, 0.000 *lgc-46(ok2949)*, 1.000 *ok2949* rescue, 0.000 *lgc-46(ok2900)*, 0.951 *ok2949* rescue. In **b** and **c**, GABAergic neuron (D-MN)-targeted rescue was achieved by using P*unc-47*. **d** Optogenetically evoked PSCs in VD5 depended on LGC-46. The experiments were performed with strains expressing channelrhodopsin-2 in cholinergic neurons (using P*unc-17*) either in the presence or absence of all-trans retinal. Compared with *wt*, the *p* values for evoked PSC amplitude and evoked PSC charge transfer are 0.000 for both RNAi and *wt* retinal (−). **e** Localization of LGC-46::GFP expressed in D-MNs and TagRFP::ELKS-1 (a presynaptic marker) expressed in cholinergic MNs in the dorsal nerve cord. The displayed images represent >20 transgenic worms. Arrows indicate co-localization of the two fusion proteins. Scale bar = 10 μm. **f** Comparison of locomotor kinematics between *wt* and the *lgc-46* RNAi strain. $p = 0.015$ Forward distance, 0.015 backward distance, 0.017 backward/forward ratio, 0.568 forward speed, and 0.844 backward speed. The asterisks indicate statistically significant differences compared with *wt* (*$p < 0.05$, **$p < 0.01$, ***$p < 0.001$) based on either one-way ANOVA with Tukey's post hoc test (**b–d**) or unpaired two-sided *t*-test (**f**). The numbers inside brackets indicate sample size (*n*). *n* = numbers of independently recorded cells in **b–d**, but numbers of individual worms in **f**. Data are presented as mean values ± SEM. Source data are provided as a Source data file.

marker (Fig. 1e). However, a significant portion of LGC-46::GFP puncta was not colocalized with or juxtaposed to TagRFP::ELKS-1, possibly due to LGC-46::GFP overexpression. Collectively, these results indicate that LGC-46 is a key postsynaptic ACh receptor mediating synaptic transmission from cholinergic MNs to D-MNs. In an earlier study, we found that LGC-46 is a key component of a postsynaptic ACh receptor in A-MNs[34]. However, the decay time constant of LGC-46-dependent spontaneous PSCs in the representative A-MN VA5 is more than 30-fold of that in VD5 (Supplementary Fig. 3), suggesting that the LGC-46 receptor in VD5 likely differs from that in VA5 in subunit compositions.

We assessed the role of synaptic transmission from cholinergic MNs to D-MNs in locomotion by comparing locomotor kinematics between wild-type and the GABAergic neuron-targeted *lgc-46* RNAi strain using *Track-A-Worm*, an automated worm-tracking system[35]. The *lgc-46* RNAi strain displayed relatively more backward but less forward movement without a change in locomotion speed (Fig. 1f). Since the direction of worm movement depends on a balance of activities between the forward and backward neural circuits[22], our results suggest that a physiological function of D-MNs is to tilt the balance in favor of forward locomotion.

**GABAergic MNs favor forward locomotion through inhibiting AVA.** How might D-MNs favor forward locomotion? D-MNs provide chemical synaptic inputs to both cholinergic MNs and AVA interneurons[27]. An inhibition of either A-MNs or AVA could indirectly favors forward locomotion. To address these two possibilities, we examined the effect of optogenetic activation of D-MNs on PSCs in VA5 using a strain expressing ChR2 specifically in GABAergic neurons[34,36]. We previously showed that spontaneous PSCs in VA5 may be divided into two types: slow and large events (sPSCs) caused by ACh release from AVA, and fast and small events (fPSCs) caused by the activation of an ACh autoreceptor[34]. Under our experimental conditions, optogenetic activation of an inotropic GABA receptor in VA5 would manifest as an evoked inward current because the Cl⁻ equilibrium potential (−6 mV) was more depolarized than the holding potential (−60 mV), whereas optogenetic inhibition of AVA would cause a reduced frequency of the sPSCs in VA5. We found that optogenetic stimulation of D-MNs did not cause any appreciable inward current but inhibited sPSCs in VA5 profoundly (>90%) and reversibly (Fig. 2a). The absence of an optogenetically evoked inward current in VA5 was unlikely due to a poor expression of ChR2 in D-MNs because blue light stimulation of the same strain causes large evoked PSCs in body-wall muscle cells[36]. Our observation is consistent with earlier

reports that D-MNs activate metabotropic rather than ionotropic GABA receptors in cholinergic MNs[37,38]. Because the *unc-47* promoter used for ChR2 expression does not have activities in any other neurons presynaptic to AVA[32], our results indicate that D-MNs likely inhibit the backward circuit through AVA.

To confirm that AVA may be inhibited by D-MNs, we examined the effects of optogenetic activation of D-MNs on AVA whole-cell current and membrane voltage in wild-type worms. We found that optogenetic stimuli caused outward current and membrane hyperpolarization (Fig. 2b), and that the GABA_A receptor blocker gabazine prevented the optogenetically evoked outward current (Fig. 2c). In contrast, the same stimulation did not cause any detectable response in AVB premotor interneurons and the B-type cholinergic NM VB6 (Supplementary Fig. 4), which is consistent with their lack of synaptic inputs from D-MNs[27]. These results suggest that D-MNs inhibit AVA through an ionotropic GABA receptor. We next set out to identify the putative GABA_A receptor in AVA. Among six GABA_A receptor genes in *C. elegans*, *unc-49* encodes a chloride channel whereas the remaining ones either encode a cation channel or are uncharacterized[39–41]. We explored the possibility of UNC-49 being the postsynaptic receptor. In *unc-49(e407)* mutant, optogenetic activation of D-MNs did not produce the inhibitory effects on either AVA or VA5 (Fig. 2a, b). AVA-targeted *unc-49* RNAi also substantially eliminated the inhibitory effects of D-MNs (Fig. 2a, b). The alleviation of the inhibitory effects of D-MNs on AVA in the *unc-49* RNAi strain was not due to a leakage of the RNAi effect into other neurons because A-MN-targeted *unc-49* RNAi did not prevent the inhibitory effect of optogentic activation of D-MNs on sPSCs in VA5 (Supplementary Fig. 5). Furthermore, we found that the frequency of inhibitory spontaneous PSCs in AVA, which appeared as upward defections in the current trace under our experimental conditions, was reduced to 8 and 23% of wild-type level in the *unc-49* mutant and RNAi strains, respectively (Fig. 2d), suggesting that D-MNs constitute a major source of inhibitory synaptic inputs to AVA. To confirm UNC-49 expression in AVA, we expressed GFP under control of the *unc-49* promoter (P*unc-49*) in a strain with AVA labeled by mStrawberry[34]. In transgenic worms, the two AVA interneurons were co-labeled by mStrawberry and GFP (Fig. 2e). Thus, our results establish UNC-49 as the postsynaptic GABA_A receptor mediating retrograde signaling from D-MNs to AVA.

We next performed several experiments to determine physiological significance of the D-AVA inhibitory circuit. First, we examined the effect of gabazine on sPSCs in VA5, and observed a concentration-dependent augmentation of sPSC frequency

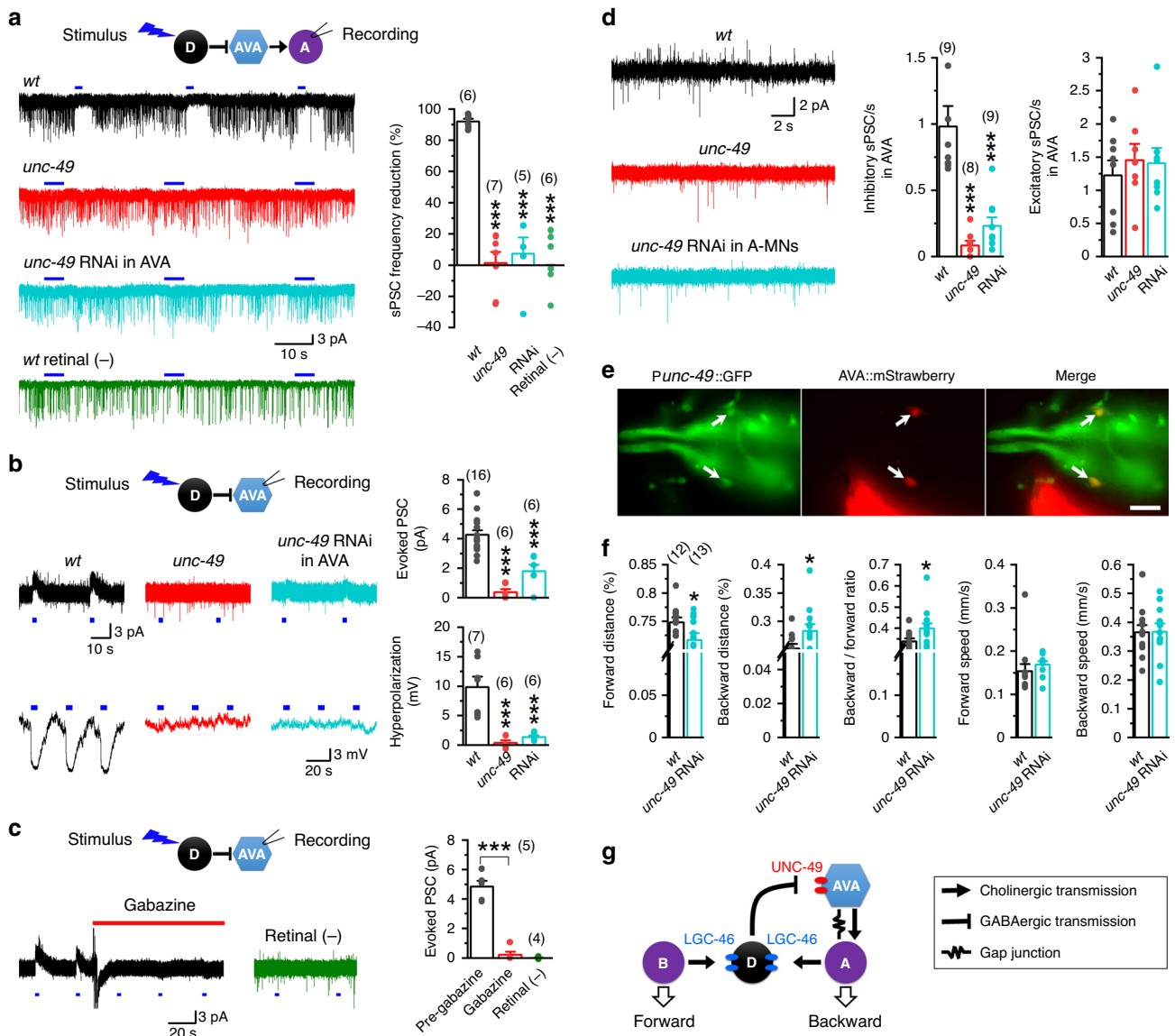

**Fig. 2 D-MNs favor forward locomotion through activating UNC-49 GABA$_A$ receptor in AVA interneurons. a** Optogenetic activation of D-MNs suppressed spontaneous postsynaptic currents (sPSCs) in VA5 (held at −60 mV) without causing an evoked inward current. All worms were treated with all-trans retinal except for those of the wild-type (*wt*) control group (indicated). The bar graph shows the percentage of sPSC reduction compared with the pre-stimulation period. Only slow and large sPSCs, which result from synaptic transmission from AVA[34], were quantified. In quantifying the effect of blue light stimulation on sPSCs, the entire interval showing an apparent lack of events was used for *wt* (with retinal) but only the interval matching the blue light pulse was used for the remaining groups. Compared with wt, $p = 0.000$ for all groups. **b** Optogenetic activation of D-MNs evoked outward current in AVA (held at −10 mV) and caused AVA hyperpolarization in *wt* but not in either *unc-49(e407)* or the AVA-targeted *unc-49* RNAi strain. Compared with *wt*, $p = 0.000$ for both groups. **c** Gabazine (500 μM) abolished AVA outward current evoked by optogenetic activation of D-MNs. Compared with Pre-gabazine, $p = 0.000$ for both groups. **d** Inhibitory sPSCs were greatly reduced in the *unc-49* mutant and RNAi strains. Compared with *wt*, $p = 0.000$ for Inhibitory sPSCs of both groups, but 0.777 and 0.844 for excitatory sPSCs of *unc-49* and RNAi, respectively. **e** Expression of GFP under the control of *unc-49* promoter in a strain with AVA neurons labeled by mStrawberry resulted in AVA colabeling by both fluorescent proteins. The displayed images represent >20 transgenic worms. Scale bar = 10 μm. **f** Comparison of locomotor kinematics between *wt* and the *unc-49* RNAi strain. $p = 0.044$ Forward distance, 0.044 backward distance, 0.049 backward/forward ratio, 0.424 forward speed, and 0.960 backward speed. **g** Diagram showing a closed-loop circuit that includes AVA, A-type cholinergic motor neurons, and D-MNs. The horizontal blue lines in **a–c** indicate the times (2 or 5 s) of blue light stimulation. The asterisks indicate significant differences (*$p < 0.05$, ***$p < 0.001$) based on either one-way ANOVA with Tukey's post hoc test (**a, b, d**), paired two-sided *t*-test (**c**), or unpaired *t*-test (**f**). The numbers inside brackets indicate sample size (*n*). *n* = numbers of independently recorded cells in **a–d**, but numbers of individual worms in **f**. Data are presented as mean values ± SEM. Source data are provided as a Source data file.

(Supplementary Fig. 6a), suggesting that the inhibitory circuit was active under our experimental conditions. Next, we assessed the effect of disrupting this circuit on PSC bursts in A-MNs because AVA controls A-MNs by producing PSC bursts[34]. Both the duration and total charge transfer of PSC bursts in VA5 were

significantly increased in the AVA-targeted *unc-49* RNAi strain (Supplementary Fig. 6b). Finally, we assessed the role of this circuit in behavior by comparing locomotor kinematics between wild-type and the AVA-targeted *unc-49* RNAi strain. Like the strain with D-MN-targeted *lgc-46* RNAi, this strain showed more

backward but less forward movement without a change in locomotion speed (Fig. 2f). Collectively, our results suggest a closed-loop system for locomotion control, in which downstream D-MNs can regulate their upstream cholinergic MNs through AVA (Fig. 2g). The inhibitory synaptic inputs are presumably from VDs because only several VDs (VD5, VD6, VD11, and VD13) are presynaptic to AVA in the worm's wiring diagram[27].

**Mechanical stimuli activate D-MNs through a stretch receptor.** We noticed in preliminary experiments that puffing any solution, including the bath solution, could cause inward current in D-MNs if the ejection pressure was sufficiently high (e.g., 10 psi), suggesting the presence of a stretch receptor. Stretch responses in worms and flies often result from the activation of degenerin/epithelial sodium channels (DEG/ENaC)[42–46], which belong to the same family of proteins as mammalian acid-sensing ion channels[47]. Because UNC-8 is the only DEG/ENaC with reported expression in D-MNs[29], we explored its potential role in the VD5 stretch response. Pressure ejection (10 psi) of the bath solution with the ejection pipette tip aimed at VD5 dendrite caused an inward current of $43.5 \pm 8.1$ pA in wild type but only $5.0 \pm 1.9$ pA in *unc-8(tm2071)*, a putative null[29]. The mutant phenotype could be rescued by expressing wild-type UNC-8 specifically in GABAergic neurons ($58.9 \pm 8.4$ pA), and recapitulated in wild-type worms by GABAergic neuron-targeted *unc-8* RNAi ($6.5 \pm 1.7$ pA) (Fig. 3a), suggesting that UNC-8 is critical to the stretch response. In contrast, VD5 whole-cell current caused by membrane voltage steps was indistinguishable between wild-type and the *unc-8* RNAi strain (Supplementary Fig. 7), suggesting that UNC-8 does not contribute to voltage-dependent whole-cell current. In addition, we determined whether VD5 stretch response is abnormal in a *del-1(ok150)* and *mec-6(u450)* double mutant because these two genes genetically interact with *unc-8* in producing a defective locomotion phenotype[29]. VD5 of the double mutant was found to have a normal stretch response (Fig. 3a), which is consistent with the absence of *del-1* and *mec-6* expression in D-MNs (www.wormbase.org). Thus, our results suggest that UNC-8 is a key stretch receptor in D-MNs.

We next determined whether stretch activation of D-MNs may inhibit AVA and impact locomotion behavior. Outward current and membrane hyperpolarization were observed in AVA of wild-type but not the AVA-targeted *unc-49* RNAi strain upon puffing (10 psi) the bath solution aimed at VD5 (Fig. 3b), indicating that the stretch stimulus was sufficient to inhibit AVA. The D-MN-targeted *unc-8* RNAi strain also showed relatively more backward but less forward locomotion without a change in locomotion speed compared with wild type (Fig. 3c). These locomotor phenotypes are similar to those of the D-MN-targeted *lgc-46* RNAi strain and the AVA-targeted *unc-49* RNAi strains. The phenotypic similarities among these strains suggest that D-MNs may favor forward locomotion by inhibiting AVA no matter they are activated by cholinergic synaptic inputs or mechanical stimuli.

**D-AVA circuit suppresses hyperosmolarity avoidance.** Because AVA interneurons act downstream of ASH sensory neurons in the avoidance response to hyperosmotic glycerol[24] (Fig. 4a), we suspected that D-MNs can regulate the avoidance response through AVA. To address this possibility, we positioned the tip of a puffing glass pipette ~20 μm away from the nose, where sensory nerve endings of ASH are located, and monitored AVA electrical properties in response to pressure (2–4 psi) ejection of either the bath solution or hyperosmotic glycerol solutions. While the bath solution had no detectable effect, glycerol caused concentration-dependent inward current and membrane depolarization in AVA (Fig. 4b and Supplementary Fig. 8), indicating that AVA can be

activated by an ASH-sensed noxious stimulus. Because AVA mediates avoidance responses through activating A-MNs, and the latter is a major source of excitatory synaptic inputs to D-MNs, we investigated whether the D-AVA circuit may modulate the avoidance response by performing three different experiments. First, we determined whether the glycerol stimulus may activate D-MNs. We observed glycerol-induced inward current and membrane depolarization in VD5 of wild-type but much less effects in that of the GABAergic neuron-targeted *lcg-46* RNAi strain (Fig. 4c), suggesting that glycerol activated VD5 through enhancing synaptic transmission from A-MNs. This conclusion is in agreement with the fact that VD5 does not receive synaptic inputs from either ASH or AVA[27]. Next, we examined the effect of disrupting the D-AVA circuit on the glycerol-induced AVA depolarization. Both the AVA-targeted *unc-49* RNAi strain and the D-MN-targeted *lgc-46* RNAi strain showed a greater degree of depolarization than wild type (Fig. 4d), suggesting that the D-AVA circuit normally antagonizes the activating effect of ASH on AVA. Finally, we assessed the effect of disrupting the D-AVA circuit on glycerol avoidance behavior by performing a hyperosmolar solution avoidance assay[17,48]. The avoidance response was induced by glycerol in a concentration-dependent manner, and was much stronger in the AVA-targeted *unc-49* RNAi strain and the D-MN-targeted *lgc-46* RNAi strain than wild type at glycerol concentrations of 2 and 3 M (Fig. 4e). However, a significant difference was not detected at 4-M glycerol, which was probably because the motivation of worms to escape was outweighed by an inhibitory effect of glycerol on locomotion. Collectively, our results suggest that a physiological function of the D-AVA circuit is to suppress hyperosmotic avoidance response.

**D-AVA circuit enhances positive chemotaxis.** AVB are downstream of AWA sensory neurons but upstream of B-MNs in the neural circuit producing positive chemotaxis to diacetyl (Fig. 5a). We investigated whether the D-AVA circuit also modulates diacetyl chemotaxis by performing three different experiments. First, we determined whether application of diacetyl to the vicinity of the nose may cause electrical changes in D-MNs. We observed PSC bursts and membrane depolarization in VD5 upon diacetyl application (Fig. 5b). These changes presumably resulted from excitatory synaptic transmission from B-MNs because they are the primary source of chemical synaptic inputs to D-MNs[18,19,27], although we cannot exclude potential minor contributions from other neurons. Next, we examined the effect of diacetyl application on AVA. We observed diacetyl-evoked outward current and membrane hyperpolarization in wild-type but not in the AVA-targeted *unc-49* RNAi strain (Fig. 5c), suggesting that diacetyl activated the D-AVA circuit. Finally, we determined whether the D-AVA circuit modulates the chemotactic response to diacetyl. We found that the chemotaxis index (CI) was substantially decreased in the AVA-targeted *unc-49* RNAi strain compared with wild type (Fig. 5d). Collectively, our results suggest that the D-AVA inhibitory circuit enhances positive chemotaxis in wild-type worms.

**AVA integrates sensorimotor inputs.** Our observations that both the glycerol avoidance and diacetyl attraction behaviors were modulated by the D-AVA circuit suggested that AVA could be a hub where sensory information from ASH and AWA circuits and motor information from D-MNs may be integrated. To address this possibility, we expressed ChR2 in ASH neurons and determined whether optogenetically evoked AVA membrane depolarization may be modulated by diacetyl and gabazine, which activates AWA and blocks the D-AVA circuit, respectively. Stimulation of ASH by blue light caused AVA depolarization, and this effect depended on

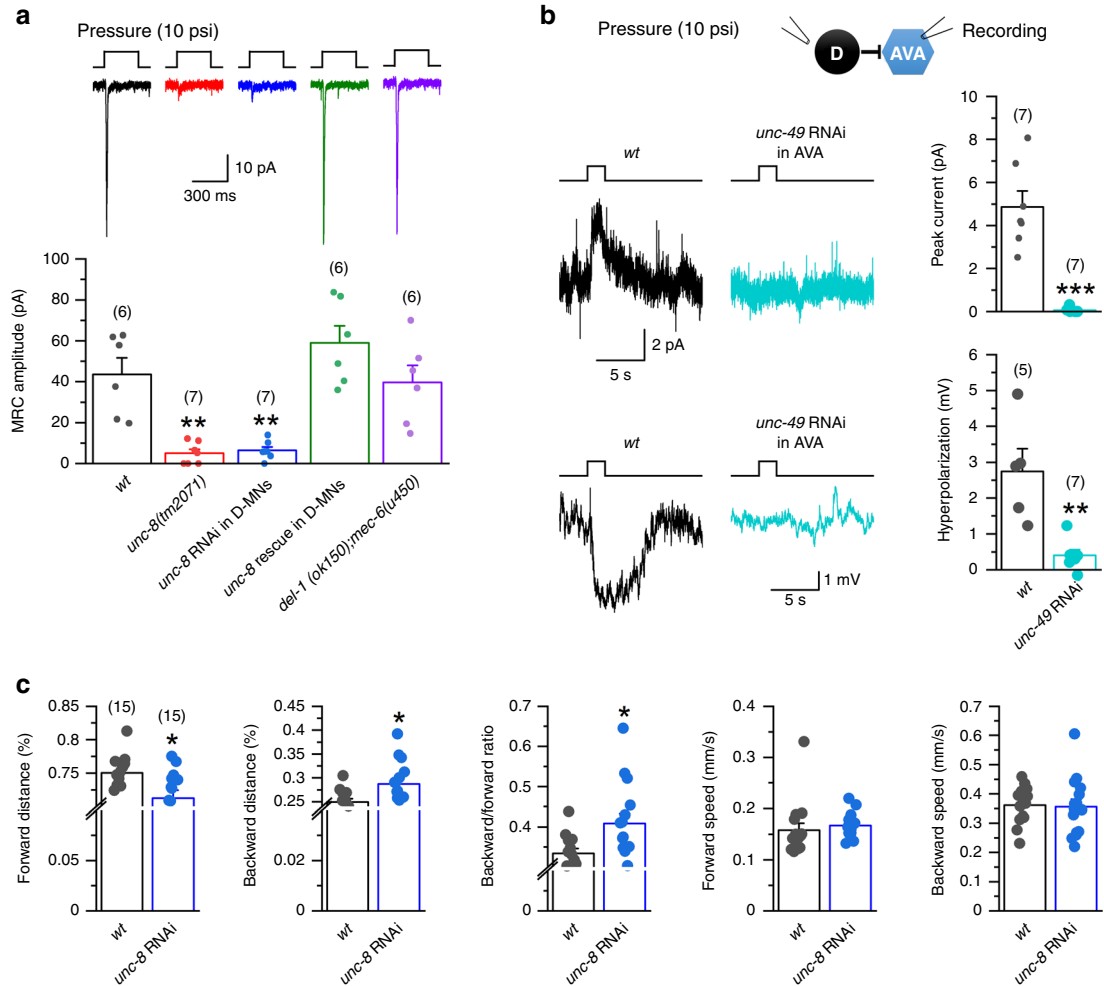

**Fig. 3 UNC-8 mediates mechanosensory responses of D-MNs and regulates locomotion through AVA interneurons. a** UNC-8 was required for mechanoreceptor current (MRC) caused by ejecting the bath solution at 10 psi in VD5 (held at −60 mV). Compared with wild type (*wt*), *p* = 0.001 *unc-8* (*tm2071*), 0.002 *unc-8* RNAi, 0.441 *unc-8* rescue, and 0.992 *del-1*(*ok150*);*mec-6*(*u450*). **b** Mechanical stimulation of D-MNs evoked an outward current in AVA and hyperpolarization of AVA in *wt* but not the AVA-targeted *unc-49* RNAi strain. Compared with *wt*, *p* = 0.000 for peak current, and 0.002 for hyperpolarization. **c** GABAergic neuron-targeted *unc-8* RNAi did not alter locomotion speed but significantly increased the ratio of backward/forward movement. *p* = 0.011 forward distance, 0.011 backward distance, 0.012 backward/forward ratio, 0.541 forward speed, and 0.876 backward speed. The asterisks indicate statistically significant differences compared with *wt* (**p* < 0.05, ***p* < 0.01, ****p* < 0.001) based on either one-way ANOVA with Tukey's post hoc test (**a**) or unpaired two-sided *t*-test (**b, c**). The numbers inside brackets indicate sample size (*n*). *n* = numbers of independently recorded cells in **a** and **b**, but numbers of individual worms in **c**. Data are presented as mean values ± SEM. Source data are provided as a Source data file.

the presence of all-trans retinal and light intensity (Fig. 6a–c). Fitting of the light intensity and membrane depolarization relationship by a Hill's equation yielded a Hill slope value of 0.87 ± 0.22 and a maximal membrane depolarization value of 9.15 ± 0.85 mV (Fig. 6d), indicating a steep relationship between light intensity and membrane depolarization and a rather small dynamic range of the depolarization, which could be the mechanism underlying hypersensitive reactions of *C. elegans* to subtle noxious stimuli.

We then examined AVA response to the optogenetic stimulation of ASH in the presence of either diacetyl or gabazine in the bath solution. We found that diacetyl reduced the slope but not the peak magnitude of AVA depolarization (Fig. 6e) whereas gabazine augmented AVA depolarization across the entire light intensity range (Fig. 6f). The gabazine effect most likely resulted from disrupting the D-AVA circuit because AVA does not receive synaptic inputs from other GABA neurons. Together, these results suggest that AVA integrates sensory information from ASH and AWA neurons, and motor information from D-MNs in a fashion of nonlinear computation.

**D-AVA circuit biases threat-reward decision-making**. The results described above indicate that AVA is activated by the glycerol repellant circuit but inhibited by the diacetyl attractant circuit. How would AVA respond when worms are confronted with glycerol and diacetyl simultaneously? To answer this question, we examined the effects of glycerol on AVA membrane voltage in the presence and absence of diacetyl. In these experiments, diacetyl was added to the bath solution whereas glycerol was puffed (2–4 psi) to the vicinity of the nose through a glass pipette (Fig. 7a). In wild type, application of glycerol for 2 s caused prolonged AVA depolarization, and this effect became much weaker in the presence of diacetyl (Fig. 7b). In contrast, diacetyl did not alter the depolarizing effect of glycerol in the AVA-targeted *unc-49* RNAi strain (Fig. 7b). These results suggest that activation of the AWA attractant circuit can mitigate the excitatory effect of the ASH repellant circuit on AVA through the D-AVA circuit.

We next determined whether the D-AVA circuit modulates decision-making when freely moving worms were confronted with both a repellant and an attractant. Specifically, we analyzed

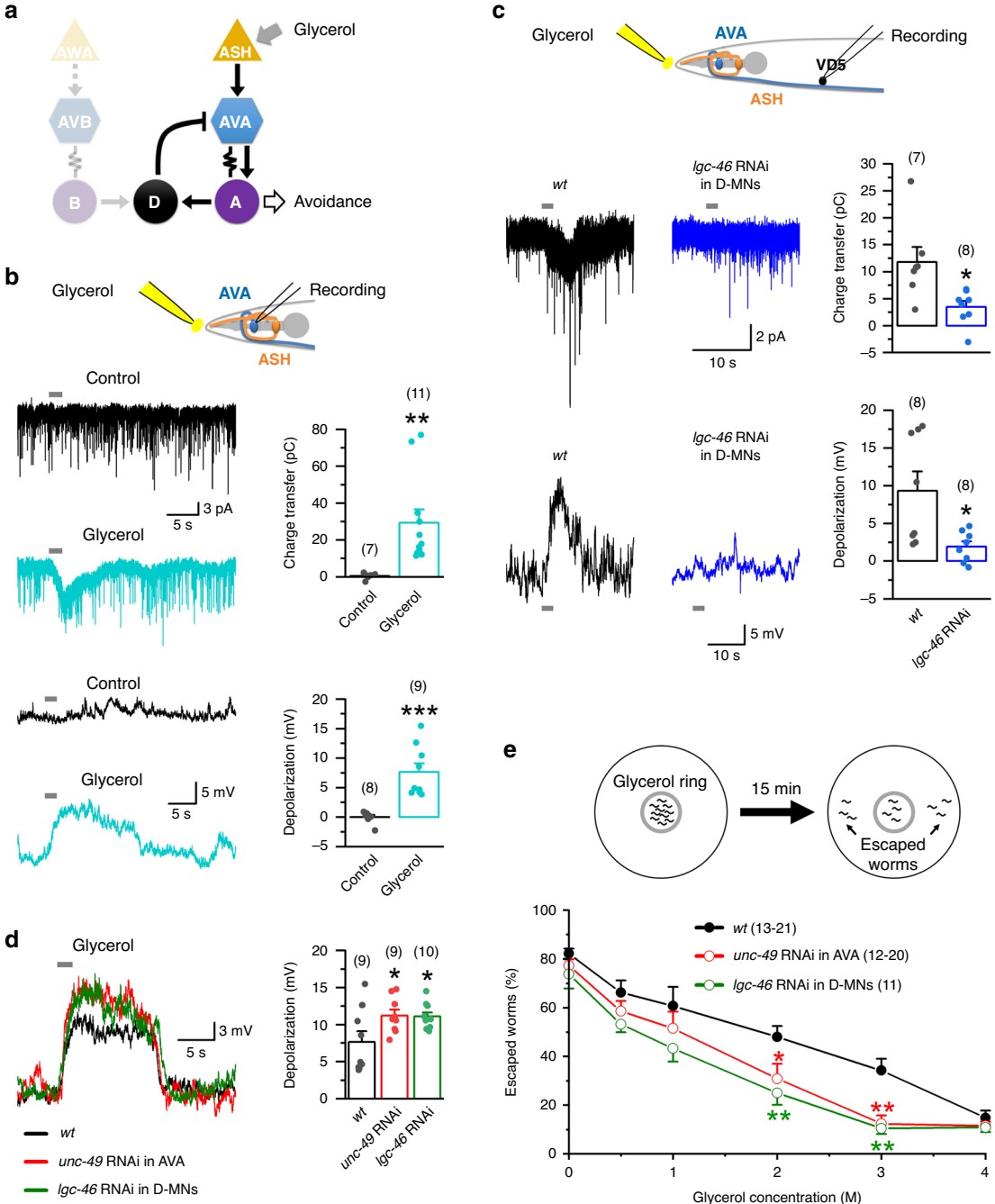

**Fig. 4 The D-AVA circuit suppresses avoidance response to noxious hyperosmolarity. a** Diagram depicting a simplified neural circuit for hyperosmolarity avoidance. **b** Application of a hyperosmotic glycerol solution but not the bath solution (control) caused inward current in AVA (held at −60 mV) and depolarization of AVA. Compared with control, $p = 0.006$ charge transfer, and 0.000 depolarization. **c** Knockdown of *lgc-46* in GABAergic neurons greatly abated glycerol-induced inward current in VD5 and depolarization of VD5 compared with wild type (*wt*). Compared with wt, $p = 0.012$ charge transfer, and 0.014 depolarization. **d, e** Knockdown of either *unc-49* in AVA or *lgc-46* in GABAergic neurons augmented glycerol-induced AVA depolarization (**d**), and reduced escape probability from a hyperosmotic glycerol barrier (**e**). In **d**, compared with *wt*, $p = 0.046$ *unc-49* RNAi, and 0.047 *lgc-46* RNAi. In **e**, 10–15 worms were placed inside a glycerol ring (diameter about 1 cm), and the percentage of worms escaped in 15 min was quantified. The effect of only one concentration of glycerol was tested in each experiment. In **b**–**d**, glycerol solution (2 M) was applied to the vicinity of the nose through a puffing pipette. The asterisks indicate significant differences (*$p < 0.05$, **$p < 0.01$, ***$p < 0.001$) compared with either the control or wild type (*wt*) based on unpaired two-sided *t*-test (**b, c**), or one-way ANOVA with Tukey's post hoc (**d, e**). The numbers inside brackets indicate sample size (*n*). $n =$ numbers of independently recorded cells in **b**–**d**, but numbers of individual worms in **e**. In **e**, sample sizes of the *wt* and *unc-49* RNAi groups varied among different glycerol concentrations. For *wt*, $n = 13, 15, 16, 21, 16,$ and 16 for the concentrations of 0–4 M. For *unc-49* RNAi, $n = 16, 20, 12, 12, 12,$ and 20 for the concentrations of 0–4 M. Data are presented as mean values ± SEM. Source data are provided as a Source data file.

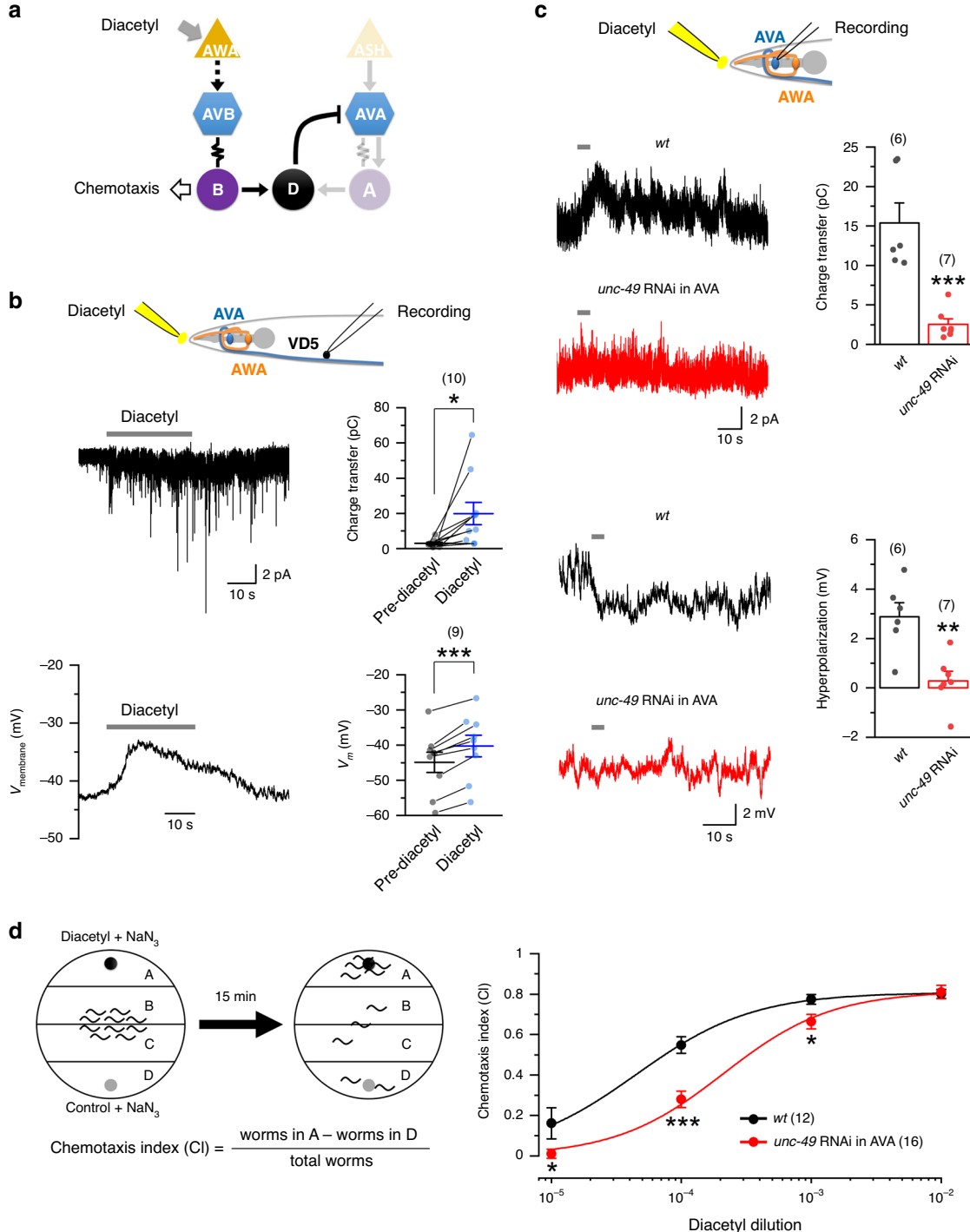

**Fig. 5 The D-AVA circuit enhances chemotaxis toward diacetyl. a** A simplified wiring diagram for diacetyl chemotaxis. **b** Diacetyl caused inward current in VD5 and depolarization of VD5. Compared with Pre-diacetyl, $p = 0.024$ charge transfer, and 0.000 $V_m$. **c** Diacetyl caused outward current in AVA and hyperpolarization of AVA in wild-type (*wt*) but not AVA-targeted *unc-49* RNAi strain. Compared with *wt*, $p = 0.000$ charge transfer, and 0.003 hyperpolarization. **d** AVA-targeted *unc-49* RNAi reduced diacetyl chemotaxis response. Left: diagram showing the chemotaxis assay. At the beginning of each assay, a drop of diacetyl and NaN$_3$ mixture and a drop of diacetyl solvent and NaN$_3$ mixture were spotted on opposite sides of a nematode culture plate (6-cm diameter), and 10–15 worms were placed in the middle of the plate. Chemotaxis index was calculated as shown. Right: diacetyl concentration and chemotaxis index curves of *wt* and AVA-targeted *unc-49* RNAi. Compared with *wt*, $p = 0.043$, 0.000, 0.023, and 0.839 at the four increasing glycerol concentrations, respectively. In **b** and **c**, the horizontal gray lines indicate the times of diacetyl application. Diacetyl (1:1000 dilution) was applied to the vicinity of the nose through a puffing pipette. Statistically significant differences from pre-diacetyl (**b**), *wt* (**c**), and *wt* at identical diacetyl concentrations (**d**) are indicated by asterisks indicate statistically significant difference (*$p < 0.05$, **$p < 0.01$, ***$p < 0.001$) compared with pre-diacetyl or *wt* based on either paired two-sided *t*-test (**b**) or unpaired two-sided *t*-test (**c, d**). The numbers inside brackets indicate sample size (*n*). $n$ = numbers of independently recorded cells in **b** and **c**, but numbers of individual worms in **d**. Data are presented as mean values ± SEM. Source data are provided as a Source data file.

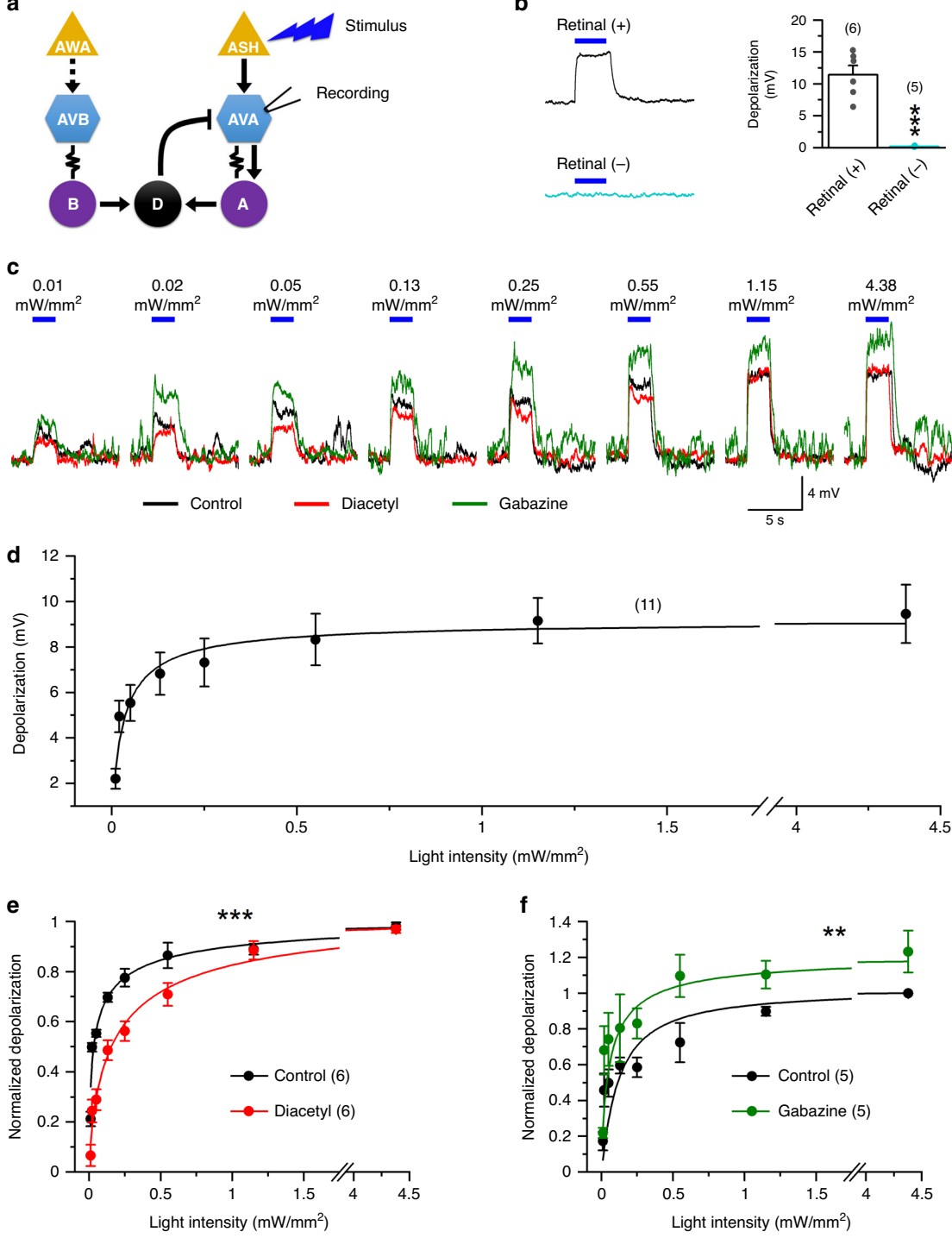

**Fig. 6 AVA interneurons are hub neurons integrating signals from diacetyl and glycerol sensory circuits and the D-AVA inhibitory circuit. a** Diagram showing the experimental approach. Optogenetically evoked AVA membrane depolarization was recorded in a strain expressing channelrhodopsin-2 specifically in ASH sensory neurons. Either diacetyl (1:1000 dilution) or gabazine (500 μM) was added to the bath solution between two series of blue light stimuli (0–4.5 mW/mm²). **b** All-trans retinal is required for optogenetically evoked AVA depolarization ($p = 0.000$). **c** Sample traces of blue light-induced AVA membrane voltage change under three different experimental conditions: control, diacetyl, and gabazine. The blue lines mark the periods of light stimuli. The numbers above them indicate light intensities (mW/mm²). **d** Light intensity and AVA membrane depolarization curve of the control, which was pooled from the diacetyl and gabazine groups. **e** Light intensity and AVA membrane depolarization curves of the gabazine group normalized to its control (pre-gabazine) peak response. $p = 0.001$ between the two groups. **f** Light intensity and AVA membrane depolarization curves of the diacetyl group normalized to its control (pre-diacetyl) peak response. $p = 0.009$ between the two groups. The asterisk in **b** indicates a significant difference compared with the Retinal (+) group whereas those in **e** and **f** indicate significant differences compared with the control (**$p < 0.01$, ***$p < 0.001$, two-way mixed design ANOVA). The numbers inside brackets indicate sample size ($n$). $n$ = numbers of independently recorded cells. Data are presented as mean values ± SEM. Source data are provided as a Source data file.

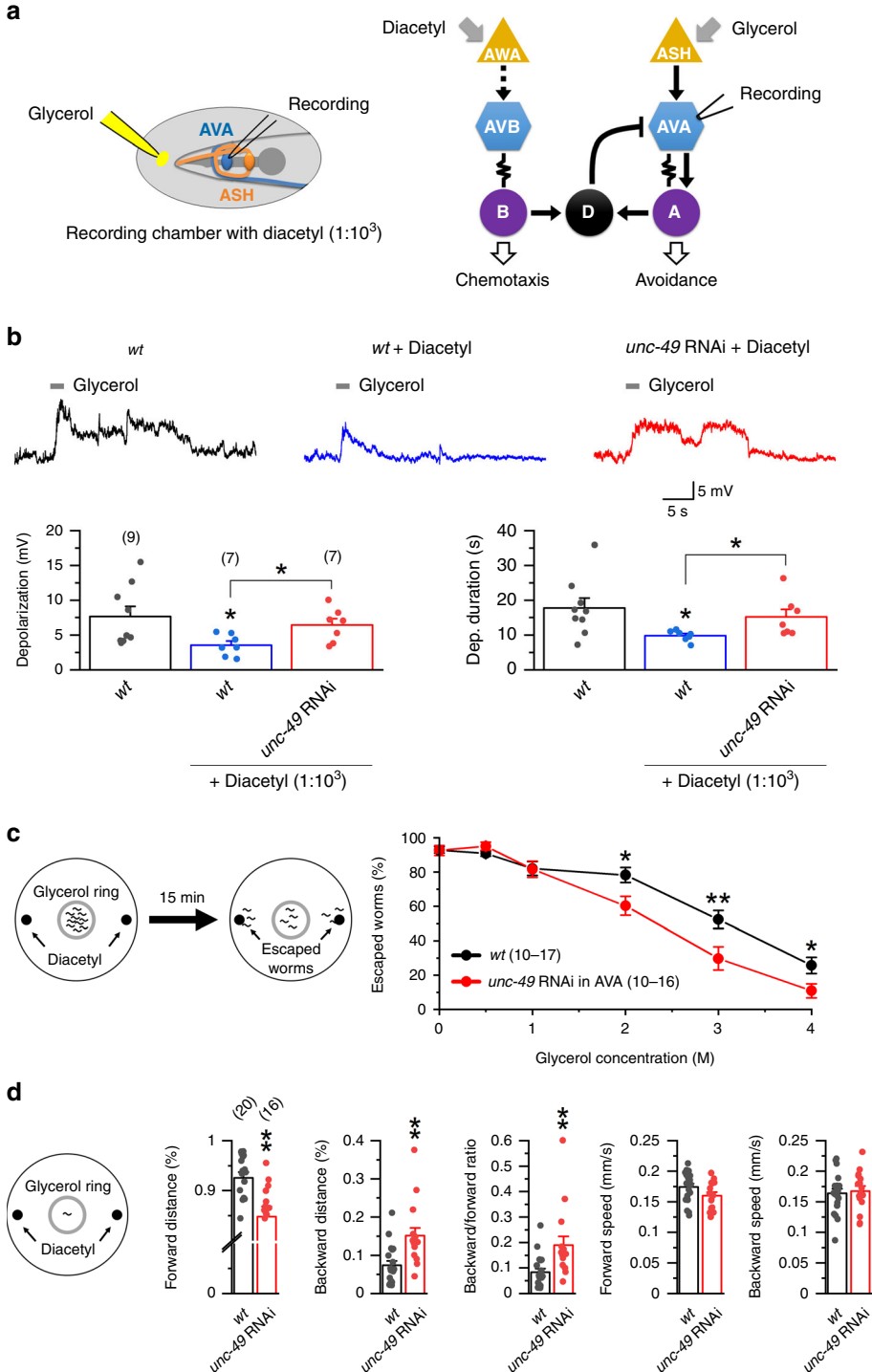

the escape response and locomotor kinematics of worms placed within a glycerol ring with two drops of diacetyl outside the ring. The percentage of worms escaped from the ring decreased with increasing concentrations of glycerol, and the escape response was substantially debilitated in the AVA-targeted *unc-49* RNAi strain compared with wild type (Fig. 7c). The AVA-targeted *unc-49* RNAi strain also showed a large decrease in forward movement but a large increase in backward movement without a change in locomotion speed (Fig. 7d). These results suggest that, when wild-type worms are confronted with both a repellent and an attractant, the D-AVA circuit may regulate decision-making by favoring the attraction behavior.

**Electrical coupling between AVA is important to decision-making.** The left and right AVA interneurons (AVAL and AVAR) are innervated by D-MNs through one synapse (from VD6) and five synapses (one from VD5, two from VD11, and two from VD13), respectively[27]. This difference suggests that the amplitude of inhibitory current caused by D-MN activation might differ between AVAL and AVAR, and that a mechanism might exist to equilibrate the inhibitory current between these two neurons. To address these possibilities, we first compared basic electrical properties between AVAL and AVAR. We observed three major differences between them: (1) AVAR was more hyperpolarized than AVAL in the resting membrane

**Fig. 7 The D-AVA circuit biases toward reward behavior in threat-reward decision-making. a** Diagram of the neural circuits mediating responses to diacetyl and hyperosmotic glycerol. **b** Glycerol-induced AVA depolarization was mitigated by diacetyl. Glycerol (2 M) was delivered to the vicinity of the worm nose whereas diacetyl was added to the bath solution (1:1000 dilution). Compared with wild type (*wt*) control, $p = 0.045$ *wt* diacetyl and 0.720 *unc-49* RNAi diacetyl for depolarization amplitude, and 0.048 *wt* diacetyl and 0.684 *unc-49* RNAi diacetyl for depolarization duration. *p* values for differences between *wt* diacetyl and *unc-49* RNAi diacetyl are 0.021 and 0.036 for depolarization amplitude and duration, respectively. **c** Disruption of the D-AVA circuit by AVA-targeted *unc-49* RNAi reduced escape probability in a multisensory behavioral assay, in which 10–15 worms were placed inside a glycerol ring (2 M, 1-cm diameter) on an agar plate (6-cm diameter) with two drops of diacetyl (1:1000 dilution) outside the glycerol ring. The percentage of worms that escaped in 15 min was quantified. Only one concentration of glycerol was tested in each assay. Compared with *wt*, $p = 0.928$, 0.170, 0.951, 0.015, 0.008, and 0.030 at glycerol concentrations of 0, 0.5, 1, 2, 3, and 4 M, respectively. **d** Locomotor kinematics of worms in the presence of both a glycerol ring and two drops of diacetyl outside the ring. $p = 0.001$ forward distance, 0.001 backward distance, 0.003 backward/forward ratio, 0.077 forward speed, and 0.724 backward speed. The asterisks indicate significant differences compared with *wt* (*$p < 0.05$, **$p < 0.01$) based on either one-way ANOVA with Tukey's post hoc test or unpaired two-sided *t*-test. The numbers inside brackets indicate sample size (*n*). $n =$ numbers of independently recorded cells in **b**, but numbers of individual worms in **c** and **d**. In **c**, sample sizes of the *wt* and *unc-49* RNAi groups varied among different glycerol concentrations. For *wt*, $n = 10$, 10, 10, 17, 14, and 11 for the concentrations of 0–4 M. For *unc-49* RNAi, $n = 13$, 13, 10, 16, 12, and 13 for the concentrations of 0–4 M. Data are presented as mean values ± SEM. Source data are provided as a Source data file.

potential (Fig. 8a), suggesting that AVAR may have a higher resting potassium conductance; (2) current injections caused larger membrane voltage changes in AVAL than AVAR (Fig. 8b), suggesting that AVAL has a higher membrane resistance than AVAR; and (3) AVAR displayed larger whole-cell current than AVAL in response to voltage steps (Fig. 8c), which is consistent with a lower membrane resistance in AVAR than AVAL. These results indicate a substantial asymmetry in biophysical properties between the two AVA interneurons.

Since gap junctions exist between AVAL and AVAR[19] (https://wormwiring.org), we determined whether they are functional by performing dual-neuron voltage-clamp recordings. In this experiment, we applied a series of membrane voltage ($V_m$) steps (−110 to +50 mV at 10-mV intervals) to one AVA interneuron (Neuron 1) from a holding potential of −30 mV while the other AVA interneuron (Neuron 2) was held constant at −30 mV to record junctional current ($I_j$). Transjunctional voltage ($V_j$) is defined as "$V_m$ of Neuron 2 − $V_m$ of Neuron 1." In response to symmetric positive and negative $V_j$ steps, bidirectionally symmetric $I_j$ was observed in both AVAL and AVAR, which gave similar $I_j - V_j$ relationships and identical gap junctional conductance ($G_j$) (Fig. 8d), indicating that AVAL and AVAR are electrically coupled through non-rectifying electrical synapses. To identify the innexin(s) responsible for the electrical coupling, we analyzed the $I_j$ between AVAL and AVAR in mutants of three innexins expressed in AVA, including UNC-7, INX-7, and UNC-9[49]. $I_j$ was normal in mutants of *unc-7* and *inx-7* but severely deficient in an *unc-9* mutant (Fig. 8e). The defective coupling of the *unc-9* mutant could be rescued by AVA-targeted expression of wild-type UNC-9, and reproduced by AVA-targeted *unc-9* RNAi in wild-type worms (Fig. 8e). These results indicate that UNC-9 plays a pivotal role in establishing the electrical coupling between AVAL and AVAR.

We next recorded AVAL and AVAR whole-cell current evoked by optogenetic activation of D-MNs. In wild-type worms, a blue light stimulus caused an outward current in both AVAL and AVAR, with AVAL/AVAR ratios of 0.74 ± 0.01 for the peak current and 0.61 ± 0.04 for the current integral (Fig. 9a). The AVAL/AVAR ratios were much smaller in *unc-9* mutant than wild type, and this phenotype could be rescued by AVA-targeted expression of wild-type UNC-9 and recapitulated by AVA-targeted *unc-9* RNAi (Fig. 9a). In addition, the total charge transfer (AVAL + AVAR) was decreased by more than 60% in *unc-9* mutant compared with wild type (Fig. 9a). These results suggest that the gap junctions not only help balance the inhibitory PSCs caused by D-MNs between AVAL and AVAR, but also amplify them.

We also tested whether excitatory synaptic inputs from ASH sensory neurons to AVA differ in functional strength and

whether such difference, if any, is equilibrated by the electrical coupling between AVAL and AVAR by analyzing the effect of optogenetic activation of ASH on evoked PSCs in AVA. AVAR is postsynaptic to ASH in 10 chemical synapses (9 from ASHR, 1 from ASHL), whereas AVAL is postsynaptic to ASH in 12 chemical synapses (all from ASHL)[19] (https://wormwiring.org). Optogenetic activation of ASH caused a similar inward current in AVAL and AVAR, and disrupting the electrical coupling between AVAL and AVAR had no effect on the evoked current (Fig. 9b). These results are not surprising because AVAL and AVAR receive a similar number of ASH synaptic inputs, and gap junctions exist between the two ASH neurons[19](https://wormwiring.org).

Lastly, we examined the role of AVA electrical coupling in threat-reward decision-making behavior (Fig. 9c). Given that the electrical coupling amplifies inhibitory inputs from D-MNs, we predicted that a disruption of the electrical coupling might have a similar effect as that of the D-AVA circuit on threat-reward behaviors. Indeed, AVA-targeted *unc-9* RNAi reduced the percentage of escaped worms in the diacetyl and glycerol assay. However, a significant difference occurred at only 3-M glycerol concentration, which is in contrast to the effect of AVA-targeted *unc-49* RNAi over a broad glycerol concentration range (Fig. 7c). The weaker effect of AVA-targeted *unc-9* RNAi on the escape probability might be due to the relatively large remaining inhibitory current (Fig. 9a).

## Discussion

This study shows that D-MNs retrogradely inhibit AVA interneurons to bias threat-reward decision-making. How can inhibition of AVA produce such an effect? In *C. elegans*, attraction and avoidance behaviors are dominated by forward and backward movements, respectively[20], and the worm's decisions to move forward or backward depend on a balance of activities between the AVB and B-MN forward circuit and the AVA and A-MN backward circuit[22]. Therefore, through inhibiting AVA activity, the D-AVA circuit may produce a bias favoring the attraction behavior by suppressing backward locomotion. In the existing model of worm locomotion neural circuit, AVA interneurons activate A-MNs, which in turn activate D-MNs[20]. Addition of the D-AVA circuit to the existing model completes a closed-loop circuit system (Fig. 2g), in which behavioral outputs from AVA may be dynamically regulated by the motor system.

Traditionally, AVA were considered as command interneurons for driving backward locomotion[23,50]. However, growing evidence suggests that AVA also play important roles in sensorimotor integration and decision-making[12]. For example, worms display variable behavioral responses to repeated presentations of

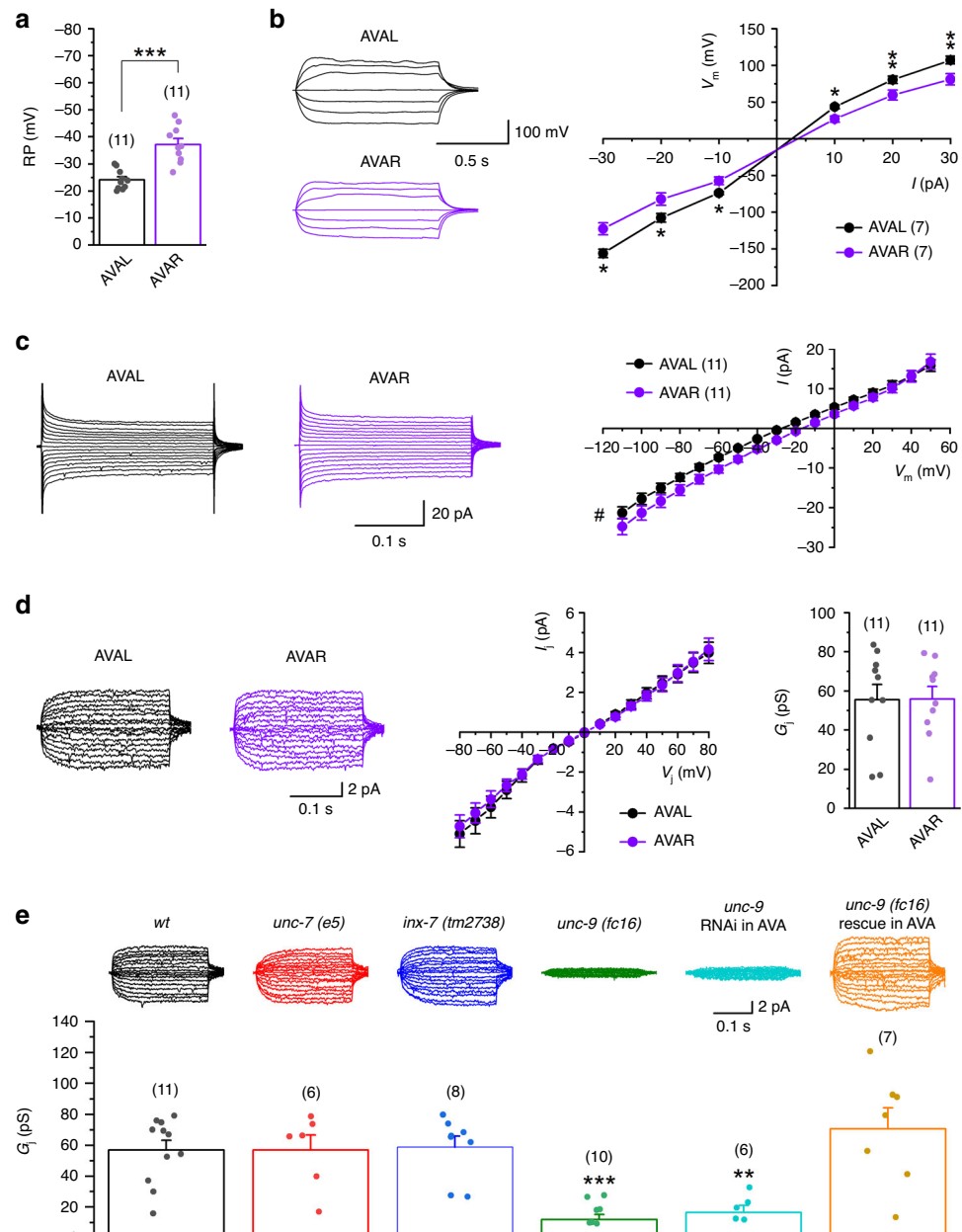

**Fig. 8 AVAL and AVAR differ in biophysical properties and are electrically coupled through UNC-9 gap junctions. a** The resting membrane potential (RP) was more hyperpolarized in AVAR (−37.2 ± 2.1 mV) than AVAL (−24.2 ± 1.1 mV). **b** Current injections caused larger membrane voltage ($V_m$) changes in AVAL than AVAR. **c** Voltage steps induced larger whole-cell current in AVAR than AVAL. The pound symbol (#) indicates significant difference between AVAL and AVAR ($p = 0.002$, two-way mixed design ANOVA). **d** Transjunctional voltage ($V_j$) steps caused junctional currents ($I_j$) between AVAL and AVAR. A series of membrane voltage steps (−110 to +50 mV) was applied to one AVA interneuron from a holding voltage of −30 mV whereas the other AVA interneuron was held constant at −30 mV to record $I_j$. Left: representative $I_j$ traces from AVAL and AVAR. Middle: $I_j − V_j$ relationships of AVAL and AVAR. Right: junctional conductance ($G_j$) based on $I_j$ from AVAL and AVAR. There is no statistically significant difference between AVAL and AVAR for both $I_j − V_j$ relationship ($p = 0.721$) and $G_j$ ($p = 0.965$). **e** Deficiencies of *unc-9* but not *unc-7* or *inx-7* inhibited the $I_j$ between AVAL and AVAR. Compared with wild type (*wt*), $p = 1.000$ *unc-7(e5)*, 1.000 *inx-7(tm2738)*, 0.000 *unc-9(fc16)*, 0.008 *unc-9* RNAi in AVA, and 0.774 *unc-9(fc16)* rescue in AVA. The asterisks indicate significant differences compared with either AVAL (**a, b**) (unpaired two-sided *t*-test) or *wt* (**e**) (one-way ANOVA with Tukey's post hoc test) (*$p < 0.05$, **$p < 0.01$, ***$p < 0.001$) whereas the pound sign (#) indicates significant difference between AVAL and AVAR (two-way mixed design ANOVA). The numbers inside brackets indicate sample size (*n*). *n* = numbers of independently recorded cells. Data are presented as mean values ± SEM. Source data are provided as a Source data file.

attractive odors such as isoamyl alcohol, and this variability is generated through interactions among AVA and two other pairs of interneurons (AIB and RIM)[26]. Our results suggest that AVA are hub neurons where sensory information from both attractant and repellant sensory modalities and motor information from D-MNs are integrated, and this property of AVA allows the motor system to participate in decision-making. This D-AVA circuit might also interact with other neurons in the worm's locomotion circuit to modulate locomotion behaviors. For example, AVA and AVB may mutually inhibit their activities through reciprocal chemical synapses between them[51]. A bilateral pair of tyraminergic RIM interneurons, which have synaptic connections

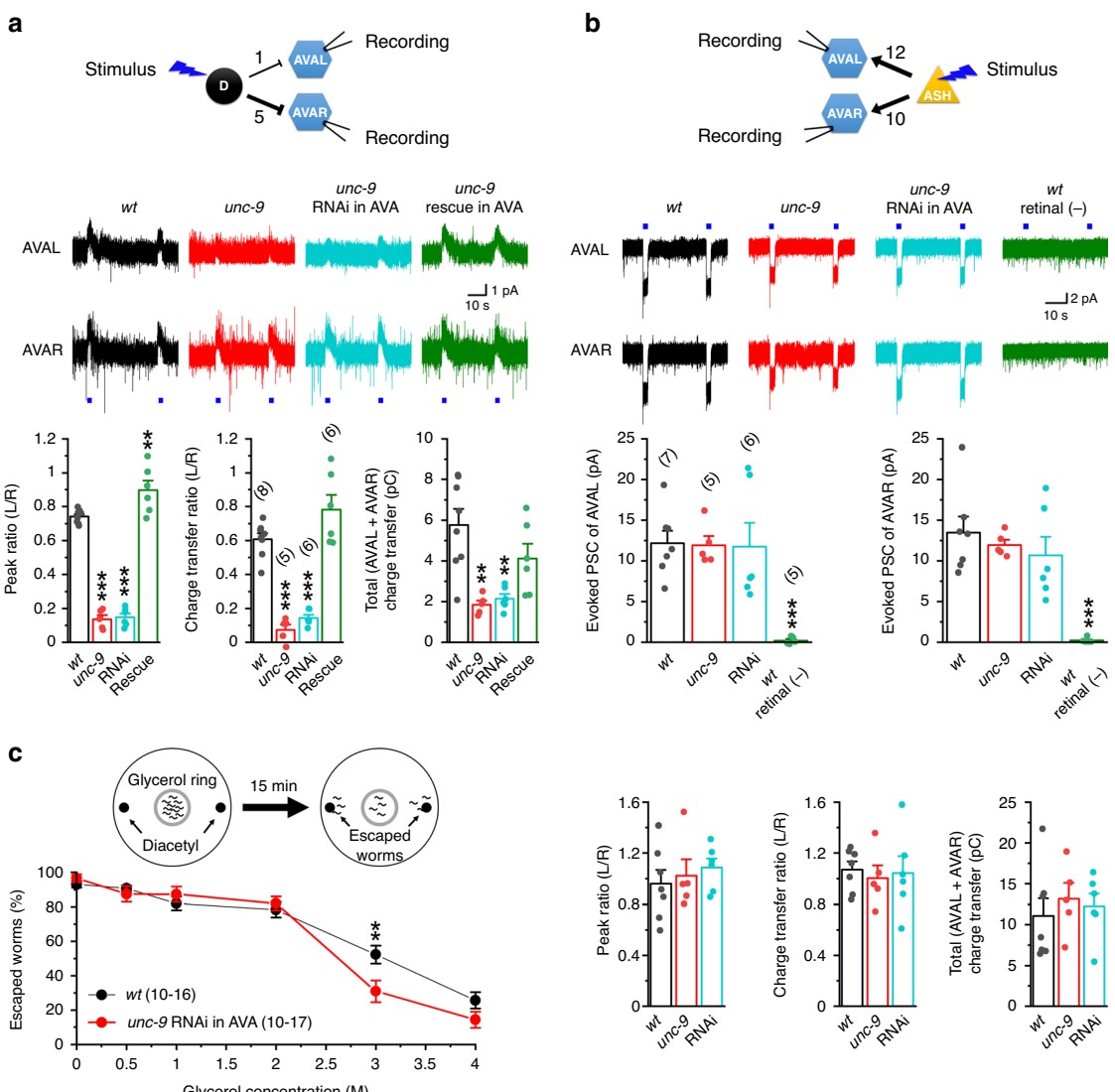

**Fig. 9 Gap junctions between AVAL and AVAR help biased threat-reward decision-making by equilibrating inhibitory synaptic inputs from D-MNs.**
**a**, **b** UNC-9 innexin is required for equilibrating inhibitory postsynaptic current (PSC) caused by optogenetic activation of D-MNs but not excitatory PSC caused by optogenetic activation of ASH sensory neurons. Top: diagram indicating the numbers of chemical synapses from D-MNs and AHS neurons to AVAL and AVAR, and the neurons for optogenetic stimulation and PSC recording. Middle: sample current traces. AVA were held at −10 mV. The blue lines indicate the times of blue light stimulation. Bottom: comparison of the AVAL/AVAR ratios of peak current and charge transfer, and the total charge transfer (AVAL + AVAR). In **a**, Compared with wild type (wt), p values of unc-9(fc16), unc-9 RNAi in AVA, and unc-9(fc16) rescue in AVA are 0.000, 0.000, and 0.008 for peak ratio, 0.000, 0.000, and 0.073 for charge transfer ratio, and 0.002, 0.002, and 0.247 for total charge transfer. In **b**, Compared with wt, the respective p values of unc-9(fc16) and unc-9 RNAi in AVA are 1.000 and 0.998 for evoked PSC of AVAL, 0.923 and 0.628 for evoked PSC of AVAR, 0.912 and 0.658 for peak ratio, 0.888 and 0.979 for charge transfer ratio, and 0.740 and 0.901 for total charge transfer. The p values between wt and wt retinal (−) were 0.000 for evoked PSC of both AVAL and AVAR. **c** Disruption of AVAL and AVAR electrical coupling reduced escape probability from a hyperosmotic glycerol barrier in a multisensory behavioral assay, in which 10–15 worms were placed inside a glycerol ring (1-cm diameter) on an agar plate (6-cm diameter) with two drops of diacetyl (1:1000 dilution) outside of the ring. The percentage of worms that escaped in 15 min was quantified. Compared with wt, p = 0.253, 0.466, 0.393, 0.544, 0.010, and 0.117 at glycerol concentrations of 0, 0.5, 1, 2, 3, and 4 M, respectively. The asterisk indicate significant differences compared with wt (**p < 0.01, ***p < 0.001) based on either one-way ANOVA with Tukey's post hoc test (**a**, **b**) or unpaired two-sided t-test (**c**). The numbers inside brackets indicate sample size (n). n = numbers of independently recorded cells in **a** and **b**, but numbers of individual worms in **c**. In **c**, sample sizes of the wt and unc-9 RNAi groups varied among different glycerol concentrations. For wt, n = 10, 10, 10, 17, 14, and 11 for the concentrations of 0–4 M. For unc-9 RNAi, n = 10, 10, 10, 16, 15, and 16 for the concentrations of 0–4 M. Data are presented as mean values ± SEM. Source data are provided as a Source data file.

with AVA and AVB, and also interact (either directly or indirectly) with primary sensory neurons[18,19,27], may modulate sensory-motor coupling and multisensory decision-making[17,24]. AVA and RIM may interact with other neurons to generate variable sensory responses[26]. Further studies are needed to determine whether and how the D-AVA circuit interacts with

other neurons to control locomotion behavior and threat-reward decision-making.

*C. elegans* body-wall muscle cells are innervated by both excitatory cholinergic MNs and inhibitory D-MNs. D-MNs innervating ventral or dorsal muscles are controlled by cholinergic MNs innervating the contralateral side. This wiring

relationship has led to the suggestion that synaptic transmission from cholinergic MNs to D-MNs allows the latter to relax antagonistic muscles on the contralateral side to help produce the sinusoidal body bends required for locomotion[20]. Worms that are either deficient in GABA release or have their D-MNs killed display a shortening of body length in response to a mechanical touch on the head[52]. Selective ablation of VDs and DDs causes ventral and dorsal navigational biases, respectively, due to somewhat deeper ventral or dorsal flexures[53]. The behavioral phenotypes observed in these studies reflect a summed effect of D-MN deficiency. It would be interesting to determine in future studies how disruptions of specific synapses with D-MNs, such as those from cholinergic MNs to D-MNs, from D-MNs to body-wall muscle cells, and from D-MNs to AVA, may affect body bending and navigational behavior.

Our results showed that the D-AVA circuit was tonically active, and modulated worm locomotion in either the absence or presence of an attractant and/or a repellant. Conceivably, the D-AVA circuit may also have tonic activities under natural conditions and play a generally important role in modulating forward vs. backward locomotion. What might be the physiological significance of a tonically active D-AVA circuit? Wild-type worms spend much more time on forward than backward movement[22]. This preference for forward movement is beneficial to worm survival because feeding occurs mainly during forward movement[24]. Given that AVA neurons are required for long reversals[50], the tonic activity of the D-AVA circuit might help suppress unnecessary long reversals to favor efficient feeding.

Left-right nervous system asymmetry is observed across animal species. For example, in humans, cortical structures for language functions are usually located in the left hemisphere whereas those for spatial recognition mainly in the right hemisphere[54]. Bilateral asymmetry is also a remarkable feature of the worm brain, with evidence mainly from two pairs of sensory neurons, AWC and ASE, which differ between the left and right in gene expression and function[55–61] (reviews[62,63]). Similarly, although the two AVA interneurons are bilaterally symmetric in gross morphology[18], this study reveals that they differ in functional properties including the resting membrane potential, membrane resistance, and strength of inhibitory synaptic inputs from D-MNs. The higher membrane resistance in AVAL than AVAR is expected to partially compensate for the weaker synaptic inputs from D-MNs to AVAL than AVAR. In addition, our results reveal that gap junctions between the two AVA neurons serve to balance the inhibitory synaptic inputs from D-MNs. This balancing act between the left and right sides of the "brain" contributes to the biased threat-reward decision-making. In mammals, a massive fiber bundle known as corpus callosum runs between the left and right hemispheres to allow both sides of the brain work together. Because there is no obvious corpus callosum-like structure in worms, gap junctions between bilateral pairs of neurons may fulfill at least some of the functions of coordinating the left and right "brains."

The two AVA interneurons also form gap junctions with some other neurons, including A-MNs. While gap junctions between AVAL and AVAR are non-rectifying UNC-9 homotypic gap junctions, those between AVA and A-MNs are strongly rectifying heterotypic gap junctions consisting of UNC-7 in AVA and UNC-9 in A-MNs[34,64]. In C. elegans, 14 of 25 innexins are expressed in neurons, and most neurons express multiple innexins and form gap junctions with several other neurons[65]. It is a standing puzzle why each neuron needs to express multiple innexins[66]. The results of this study suggest that UNC-7 and UNC-9 in AVA do not associate with innexins in other neurons indiscriminately but rather in a cell-specific manner to form molecularly and functionally different gap junctions.

In mammals, proprioceptors in muscles (muscle spindles and Golgi tendon organs) play important roles in muscle function and locomotion. Sensory signals from these proprioceptors are conducted by specific neural fiber tracks to the cerebellum to coordinate movement. However, C. elegans body-wall muscles do not have structures like the mammalian proprioceptors. It was speculated many years ago that MNs may have proprioceptive properties[18]. This possibility was confirmed by a recent study showing that C. elegans B-MNs are proprioceptive, and that this property depends on a stretch receptor[67] although molecular identity of the putative stretch receptor remains mysterious. Our results indicate that stretch-dependent modulation of locomotion behavior is not a property unique to B-MNs, and that the D-AVA circuit may be activated by both chemical synaptic transmission from cholinergic MNs and activation of the UNC-8 stretch receptor in D-MNs. Under physiological conditions, the UNC-8 stretch receptor in D-MNs is likely activated by worm body bending, like the putative stretch receptor in B-type cholinergic MNs[67]. Thus, D-MNs might integrate excitatory synaptic signals from cholinergic MNs and UNC-8 stretch receptor-sensed body bending information to modulate locomotion behavior and threat-reward decision-making through the D-AVA circuit.

An earlier study identified ACR-12 as a postsynaptic receptor in D-MNs that mediates synaptic transmission from cholinergic MNs[28]. This function of ACR-12 was concluded based on the observations that acr-12 mutants showed an increased sensitivity to an immobilizing effect of aldicarb (a cholinesterase inhibitor), a reduced frequency of spontaneous PSCs recorded from body-wall muscle cells, an abnormal locomotion waveform, and partial co-localization between a synaptic vesicle marker expressed in cholinergic MNs and GFP-tagged ACR-12 expressed in D-MNs[28]. Consistently, subsequent studies showed that ACR-12 is localized to spine-like structures in D-MNs[68–70]. Unexpectedly, our direct recoding of spontaneous PSCs from D-MNs showed that ACR-12 does not seem to play a role in mediating synaptic transmission from cholinergic MNs. Instead, we found that LGC-46 performs this function in D-MNs, based on the analysis of spontaneous and optogenetically evoked PSCs in D-MNs. Nevertheless, we cannot definitely exclude the reported role of ACR-12. In our dissected worm preparations, some of the laterally projecting commissures that allow synaptic interactions between cholinergic MNs and D-MNs were disrupted, which could have restricted our detection of spontaneous PSCs to a subset of the synaptic events. Further studies are needed to resolve the reported function of ACR-12 in the synaptic transmission from cholinergic MNs to D-MNs.

We previously showed that LGC-46 functions as a postsynaptic receptor in A-MNs mediating synaptic transmission from AVA[34]. This function and its newly identified function in D-MNs are presumably due to conduction of inward current. However, it has also been reported that LGC-46 may function as an anion channel at presynaptic sites of cholinergic MNs to inhibit neurotransmitter release[71]. The different functions of LGC-46 receptors likely result from different molecular compositions of heteromeric receptors. There is evidence suggesting that the LGC-46 receptor at presynaptic sites in cholinergic MNs also contains ACC-4 as a component[71]. Although LGC-46 alone may form a homomeric ACh receptor in the Xenopus oocyte heterologous expression system[34], the 50-fold difference in the decay time constants between LGC-46-mediated spontaneous PSCs in VA5 and VD5 (Supplementary Fig. 3) suggests that LGC-46 likely co-assembles with different proteins in these two types of MNs to form functionally distinct receptors.

In mammals, GABAergic neurons have been implicated in decision-making. For example, disrupting GABAergic transmission in insular cortex or media prefrontal cortex in rat can alter or impair decision-making[72,73]. However, it is difficult to investigate

**Table 1 List of worm strains.**

| Strain ID | Genotype |
|---|---|
| N2 | Wild-type (Bristol strain) |
| RB2155 | lgc-46(ok2900) |
| VC2209 | lgc-46(ok2949) |
| ZW795 | zwEx186[Punc-47::lgc-46ss(wp1468), Punc-47::lgc-46as(wp1469), Pmyo-2::YFP(wp214)] |
| ZW793 | lgc-46(ok2949); zwEx185[Punc-47::lgc-46(wp1506), Pmyo-2::YFP(wp214)] |
| ZW794 | lgc-46(ok2900); zwEx185[Punc-47::lgc-46(wp1506), Pmyo-2::YFP(wp214)] |
| EG5182 | oxIs407[Punc-17::ChR2::mCherry, lin-15(+)] |
| ZW1411 | oxIs407[Punc-17::ChR2::mCherry, lin-15(+)];zwEx186[Punc-47::lgc-46ss(wp1468), Punc-47::lgc-46as(wp1469), Pmyo-2::YFP(wp214)] |
| ZW1193 | zwEx254[Punc-17(delta)::unc-17ss(wp1777), Punc-17(delta)::unc-17as(wp1778), Pacr-5::GFP(wp1768)] |
| RB1559 | acr-2(ok1887) |
| VC649 | acr-9(ok933) |
| VC188 | acr-12(ok367) |
| RB1132 | acr-14(ok1155) |
| RB1226 | acr-18(ok1285) |
| RB1502 | lgc-26(ok1770) |
| MT14678 | lgc-40(n4545) |
| RB918 | acr-16(ok789) |
| CB1072 | unc-29(e1072) |
| VC2937 | unc-38(ok2896) |
| ZZ37 | unc-63(x37) |
| EG5025 | oxIs351[Punc-47::ChR2::mCherry, lin-15(+)] |
| ZW754 | oxIs351[Punc-47::ChR2::mCherry, lin-15(+)]; zwEx175[Pflp-18::loxP::LacZ::STOP::loxP::mCherry::SL2::GFP(wp1383), Pgpa-14::Cre(wp1339)] |
| ZW759 | unc-49(e407);oxIs351[Punc-47::ChR2::mCherry, lin-15(+)];zwEx175[Pflp-18::loxP::LacZ::STOP::loxP::mCherry::SL2::GFP(wp1383), Pgpa-14::Cre(wp1339)] |
| ZW 1176 | oxIs351[Punc-47::ChR2::mCherry, lin-15(+)];zwEx255[Pflp-18::loxP::LacZ::STOP::loxP::unc-49ss(wp1441), Pflp-18::loxP::LacZ::STOP::loxP::unc-49as(wp1442), Pflp-18::loxP::LacZ::STOP::loxP::mStrawberry(wp1392), Pgpa-14::Cre(wp1339), lin-15(+)] |
| ZW799 | zwEx187[Punc-49::GFP(wp1427), Pflp-18::loxP::LacZ::STOP::loxP::mStrawberry(wp1392), Pgpa-14::Cre(wp1339), lin-15(+)] |
| ZW1139 | zwEx255[Pflp-18::loxP::LacZ::STOP::loxP::unc-49ss(wp1441), Pflp-18::loxP::LacZ::STOP::loxP::unc-49as(wp1442), Pflp-18::loxP::LacZ::STOP::loxP::mStrawberry(wp1392), Pgpa-14::Cre(wp1339), lin-15(+)] |
| ZW1413 | oxIs351[Punc-47::ChR2::mCherry, lin-15(+)];zwIs142[Psra-11::GFP(wp712)] |
| ZW816 | zwEx192[Punc-4::unc-49ss(wp1570), Punc-4::unc-49as(wp1571), Pmyo-3::mCherry(wp756)] |
| ZW1081 | del-1(ok150);mec-6(u450) |
| ZW1013 | unc-8(tm2071) |
| NC2601 | Is[Pttr-39::unc-8ss(pSA76), Pttr-39::unc-8as(pSA78), Pttr-39::mCherry, unc-119(+)][80] |
| ZW1376 | unc-8(tm2071);zwEx256[Punc-25::unc-8::GFP(pTWM62), Pflp-18::loxP::LacZ::STOP::loxP::mCherry::SL2::GFP(wp1383), Pgpa-14::Cre(wp1339)] |
| ZW1140 | zwIs143[Pflp-18::loxP::LacZ::STOP::loxP::mStrawberry(wp1392), Pgpa-14::Cre(wp1339), lin-15(+)] |
| ZW1371 | zwEx257[Psra-6::ChR2(wp1877), Pflp-18::loxP::LacZ::STOP::loxP::mStrawberry(wp1392), Pgpa-14::Cre(wp1339), lin-15(+)] |
| CB5 | unc-7(e5) |
| FX02738 | inx-7(tm2738) |
| CW129 | unc-9(fc16) |
| ZW1238 | zwEx258[Pflp-18::loxP::LacZ::STOP::loxP::unc-9ss(wp1793), Pflp-18::loxP::LacZ::STOP::loxP::unc-9as(wp1794), Pflp-18::loxP::LacZ::STOP::loxP::mStrawberry(wp1392), Pgpa-14::Cre(wp1339)] |
| ZW1227 | unc-9(fc16); oxIs351[Punc-47::ChR2::mCherry, lin-15(+)];zwEx259[Pflp-18::loxP::LacZ::STOP::loxP::mCherry::SL2::GFP(wp1383), Pflp-18::loxP::LacZ::STOP::loxP::mCherry::SL2::unc-9(wp1813), Pflp-18::loxP::LacZ::STOP::loxP::mStrawberry(wp1392), Pgpa-14::Cre(wp1339)] |
| ZW1232 | oxIs351[Punc-47::ChR2::mCherry, lin-15(+)];zwEx258 [Pflp-18::loxP::LacZ::STOP::loxP::unc-9ss(wp1793), Pflp-18::loxP::LacZ::STOP::loxP::unc-9as(wp1794), Pflp-18::loxP::LacZ::STOP::loxP::mStrawberry(wp1392), Pgpa-14::Cre(wp1339)] |
| ZW1386 | unc-9(fc16);zwEx257[Psra-6::ChR2(wp1877), Pflp-18::loxP::LacZ::STOP::loxP::mStrawberry(wp1392), Pgpa-14::Cre(wp1339), lin-15(+)] |
| ZW1372 | zwEx257[Psra-6::ChR2(wp1877), Pflp-18::loxP::LacZ::STOP::loxP::mStrawberry(wp1392), Pgpa-14::Cre(wp1339), lin-15(+)];zwEx260[Pflp-18::loxP::LacZ::STOP::loxP::unc-9ss(wp1793), Pflp-18::loxP::LacZ::STOP::loxP::unc-9as(wp1794), Pflp-18::loxP::LacZ::STOP::loxP::mCherry::SL2::GFP(wp1383), Pgpa-14::Cre(wp1339)] |
| ZW1569 | zwEx285[Punc-47::lgc-46::GFP (wp1657), Punc-17::TagRFP::ELKS-1(wp1676), lin-15(+)]; lin-15(n765) |

whether retrograde signaling from them contributes to this function for technical reasons. This study shows that D-MNs can retrogradely modulate threat-reward decision-making by acting as a major component in a closed-loop neural circuit. Given that organizing principles of neural circuits are often conserved[74–79], the findings of this study may provide insights about how GABAergic neurons may interact with other neurons to modulate decision-making in other systems, including mammals.

## Methods

**C. elegans culture and strains.** All worms were raised on Nematode Growth Medium (NGM) agar plates seeded with OP50 *Escherichia coli* at 21 °C inside an environmental chamber. The worm strains used in this study are listed in Table 1.

**Gene expression pattern analyses.** To test whether *unc-49* is expressed in AVA interneurons, we coinjected four plasmids, wp1427 (*Punc-49::GFP*), wp1339 (*Pgpa-14::Cre*), wp1392 (*Pflp-18::loxP::LacZ::STOP::loxP::mStrawberry*), and *lin-15(+)* into the *lin-15(n765)* strain. The expression of wp1339 and wp1392 results in mStrawberry labeling of the two AVA interneurons[34,81]. The expression patterns of GFP and mStrawberry were imaged with an inverted microscope (TE-2000U, Nikon) enhanced GFP/fluorescein isothiocyanate and mCherry/Texas Red filter sets (49002 and 49008, Chroma Technology Corporation, Rockingham, VT, USA) and a Hamamatsu ORCA-Flash4.0 digital camera (Model C11440-22CU). *Pflp-18* and *Pgpa-14* were gifts from Dr Alexander Gottschalk[81] whereas *Punc-49* was cloned from genomic DNA of the Bristol N2 strain by PCR using primers listed in Supplementary Table 1.

**RNA interference.** Cell-specific RNAi was achieved by expressing plasmids encoding complementary sense and antisense mRNA fragments under specific promoters. *Punc-4* and *Punc-47* were used for expression in A-MNs and D-MNs,

respectively, whereas *Pgpa-14* and *Pflp-18* were used for AVA-specific expression. The primers for amplifying the target mRNA fragments in the sense direction were listed in Supplementary Table 1.

**Electrophysiology**. All electrophysiological experiments were performed with young adult hermaphrodites. Briefly, a worm was immobilized on a glass coverslip by applying Vetbond Tissue Adhesive (3M Company, St Paul, MN) in a drop of the bath solution. Application of the glue was restricted to the dorsal side of either the head region (for recording from head neurons) or the mid-anterior portion (for recording from motor neurons) of the worm, which allowed either the tail or both the head and tail to sway freely during the experiment. A short (200–300 μm) longitudinal incision was made by cutting through the glued portion using a diamond dissecting tool (72028, Electron Microscopy Sciences, Hatfield, PA, USA) to expose neurons of interest. The cuticle flap was folded back and glued to the coverslip. The dissected worm preparation was treated with collagenase A (Roche Applied Science, catalogue number 10103578001, 0.5 mg/ml) for 10–15 s before being flushed away by five- to tenfold of the bath solution. Borosilicate glass pipettes (tip resistance ~20 MΩ) were used as electrodes. Motor neurons were identified based on their anatomical locations interneurons based on labeling by a fluorescent protein. Whole-cell current- and voltage-clamp recordings were performed on a Nikon FN1 microscope with 4× and 40× objectives with a Multiclamp 700B amplifier (Molecular Devices, Sunnyvale, CA, USA) and Clampex software (version 10, Molecular Devices) on pressure ejections of ACh, glycerol, diacetyl, and the control solution were done with a Picospritzer III microinjector (Parker Hannifin, Hollis, NH) connected to a glass pipette with a tip diameter of ~2 μm at 2–4 psi. Mechanical activation of stretch receptors was achieved by puffing the bath solution using the Picospritzer microinjection at 10 psi. The bath solution contained (in mM) NaCl 140, KCl 5, CaCl$_2$ 5, MgCl$_2$ 5, dextrose 11, and HEPES 5 (pH 7.2). The pipette solution contained (in mM) 120 KCl, 20 KOH, 5 Tris, 0.25 CaCl$_2$, 4 MgCl$_2$, 36 sucrose, 5 EGTA, and 4 Na$_2$ATP (pH 7.2) except for experiments of recording outward whole-cell current, in which the pipette solution contained (in mM) 6.8 KCl, 113.2 Kgluconate, 20 KOH, 5 Tris, 0.25 CaCl$_2$, 4 MgCl$_2$, 36 sucrose, 5 EGTA, and 4 Na$_2$ATP (pH 7.2). All current-clamp recordings were performed without current injection.

**Optogenetic stimulation**. Worms expressing ChR2 first grew to L1-L2 stage on standard NGM plates. They were then transferred to new NGM plates either with or without (for negative control) all-trans retinal 2 days before experiments. The retinal plates were prepared by spotting each plate (60-mm diameter with 10-ml agar) with 200-μl OP50 containing 2-mM retinal (R2500, Sigma-Aldrich). In most experiments, blue light pulses (2 sec or 5 sec, 470 ± 20 nm) were generated by a Lambda XL light source (Sutter Instrument, Novato, CA, USA) with a 470 ± 20-nm excitation filter (59222, Chroma Technology Corp., Bellows Falls, VT, USA) and SmartShutter® (Sutter Instrument). Light intensity was adjusted by applying the three standard neural density filters (ND4, ND8, ND16) of the Nikon FN1 microscope in various combinations, which resulted in eight different light intensities ranging from 0.01 to 4.38 mW/mm$^2$. The maximal light intensity was used in all experiments except those assessing the effect of light intensity-dependent activation of ASH neurons on AVA membrane potential. Light intensities were measured with an optical power meter (PM100A, Thorlabs, Newton, NJ, USA) equipped with a photodiode power sensor (S121C, Thorlabs). The on-and-off of light stimulation was controlled by NIS-Elements imaging software (version 4.51) through the SmartShutter in the Lambda XL light source.

**Behavioral assays**. All behavioral assays were performed with young adult hermaphrodites on 6-cm diameter NGM plates without food. To quantify locomotion behavior, a single worm was transferred to the center of a NGM plate. After a 30-s recovery period from the transfer procedure, the worm was imaged for 1 min at 15 frames/s. Both imaging acquisition and subsequent quantitative analyses were performed using an automated worm-tracking system, *Track-A-Worm*[35].

Osmotic avoidance and multisensory assays were performed based on established procedures[17,48]. Briefly, a glycerol ring was created by applying 10-μl glycerol solution to the surface of a NGM plate by tracing a circle (1 cm in diameter) drawn on its back. In multisensory assays, two drops of 1-μl diluted diacetyl (1:1000 in water) were added to opposite sides (near the edge) of the NGM plate outside of the glycerol ring. After a 5-min equilibration period, worms were transferred to the center of the glycerol ring. In this procedure, 50–100 young adult worms were first transferred from a standard NGM plate with food to a NGM plate without food. After adding M9 buffer (100 μl) onto the plate, ~20 μl of the added M9 solution (usually containing 10–15 worms) was pipetted into the glycerol ring. Worms outside and inside the glycerol ring were counted 15 min later.

Glycerol chemotaxis assay was performed following similar procedures of previous studies[14,17]. Two parallel lines were drawn on the back of a NGM plate to divide the plate into four regions (A, B, C, and D) of an equal height. One microliter diluted diacetyl (1:1000) and 1-μl water were applied to regions A and D, respectively, followed by adding 1-μl NaN$_3$ (0.1 M) to both regions. About ten worms were transferred to the center of the plate using the procedures described above. Worms in the various regions were counted 15 min later to calculate the CI:
$$CI = \frac{\text{Worms in A} - \text{Worms in D}}{\text{Total number of worms}}.$$

**Chemicals**. Diacetyl (Sigma-Aldrich) was diluted in water to reach the 1:1000 ratio. Glycerol (BP229-1, Fisher Scientific) was diluted to final concentrations in a buffer containing 100-mM NaCl, 10-mM KCl, and 30-mM Tris [PH 7.5]. ACh (AC159170050, ACROS Organics) and gabazine (S106, Sigma-Aldrich) were first dissolved in water to make aliquots of 10-mM frozen stocks, which were diluted to final concentrations using the bath solution before use.

**Data analyses**. Frequencies and amplitudes of spontaneous PSCs were quantified using MiniAnalysis (Synaptosoft, Inc., Decatur, GA). For spontaneous PSCs in VA5, only the slow and large events (sPSCs), which were identified based on thresholds of amplitude (>5 pA) and decay time (>5 ms), were used for statistical analyses. The resting membrane potential and PSCs caused by glycerol, diacetyl, optogenetic and mechanical stimuli, and exogenous ACh were quantified with Clampfit (version 10, Molecular Devices). The duration and charge transfer of PSC bursts were quantified with Clampfit, while the frequency of PSC bursts was manually counted. Amplitudes of whole-cell current in response to voltage steps were determined from the mean current during the last 100 ms of the 1.2-s voltage steps using the Clampfit. Junctional current and membrane voltage were quantified by measuring the mean amplitude during the last 100 ms of transjunctional voltage steps and current injection steps, respectively, using Clampfit.

Data graphing were performed with Origin 2019 (OriginLab Corporation, Northampton, MA). Statistical analyses with performed with either Origin or SPSS (IBM Corporation, New York, USA). All data are shown as mean values ± SEM. Either ANOVA (one-way or two-way) or *t*-test (paired or unpaired) was used for statistical comparisons as specified in figure legends.

**Reporting summary**. Further information on research design is available in the Nature Research Reporting Summary linked to this article.

## Data availability

This work did not include any data which mandated deposition in public databases. Source data are provided with this paper.

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

## Acknowledgements

This work was supported by National Institute of Health (NS109388, MH085927 to Z.-W.W.; GM113004 to B.C.). We thank Piali Sengupta for helpful suggestions about the manuscript, Michael Nonet for a TagRFP::ELKS-1 plamid, Miriam Goodman for *unc-8 (tm2071)*, David Miller for NC2601 strain and P*unc-25::unc-8::GFP*(pTWM62) plasmid, and Caenorhabditis Genetics Center (USA) and National BioResource Project (Japan) for other mutant strains.

## Author contributions

P.L. designed and performed experiments, analyzed data, and wrote the manuscript. B.C. performed experiments. Z.-W.W. supervised the project and edited the manuscript.

## Competing interests

The authors declare no competing interests.
