## [Peer Review File · Nature Communications]

Reviewers' comments:

Reviewer #1 (Remarks to the Author):

The authors aim to show that D-MNs are able to bias the worm's behavior toward reward-decision making by preventing avoidance in threat response. Specifically, they show that D-MNs inhibit upstream AVA via a GABA receptor, UNC-49. AVAL and R differ electrically and balance this inhibition to arrive at a decision. D-MNs receive two excitatory inputs: cholinergic MNs & current through a stretch receptor. They also identify receptors on D-MNs which mediate cholinergic MN synapses, and identify receptors on AVA mediating D-MN inhibition of AVA and the innexin responsible for coupling AVA-L & AVA-R. I do have a few concerns

Major

1. A previous study Petrash et al J neurosci 33, 5524-5532 (2013) identify acr-12 as a post-synaptic receptor for D-MNs for cholinergic MN synapses. How do the authors reconcile their results with this data?
2. The authors use optogenetic stimulation of GABAergic neurons and suggest that D-MNs are responsible? Aren't there additional GABA synapses on AVA?
3. Is the D-MN effect on AVA present in all instances of forward vs backward locomotion or is it specific to aversion/reward stimuli?
4. The authors stimulate ASH, which also has uneven inputs to AVAL and R, but do not observe the same differences on either side. These results should be discussed.
5. The authors the effect of gabazine on spontaneous activity in VA5 (downstream of D-MNs through AVA) and found that sPSCs were increased in a dosaged dependent way in VA5 so this is an active circuit at rest (disinhibited AVA, leading to excitation of VA5). This could also be due to other GABAergic inputs on VA5.
6. The authors use a ring avoidance assay to prove the role of D-MN-AVA synapses. In this assay, we expect that when D-AVA circuit is disrupted in unc-49 RNAi, that we see an increase in backward locomotion. Forward locomotion we assume is necessary for escape. So, if D-AVA promotes forward locomotion then we should have more escaping worms. This appears to be true at intermediate glycerol concentrations, but not high?

Minor

1. Supplemental Figure 4 is mis-written as Supplemental Figure 3 in the text.
2. Figure 6 legend mentions "Depolarization last time of AVA", which is unclear.
3. Reference for AVAL has 1 D-MN synapse, and AVAR has 5 D-MN synapses

Reviewer #2 (Remarks to the Author):

The integration of sensory inputs to make appropriate behavioral responses is well-known to be essential for organismal survival, but evidence that motor systems actively participate in shaping sensory responsiveness has only recently begun to emerge. Here Liu et al. use electrophysiology and genetic approaches to address questions about the circuit and molecular mechanisms that regulate responses to aversive and attractive stimuli in the nematode *C. elegans*. The authors provide evidence that D-class GABAergic motor neurons inhibit the backward locomotory circuit through inhibitory synaptic contacts onto AVA locomotory control interneurons that have primary roles in promoting backward movement. The authors use AVA-specific RNAi to knockdown expression of the GABAA receptor UNC-49 to show to argue that synaptic inhibition of AVA from GABAergic motor neurons biases animals toward forward locomotion during threat-reward decision making. In particular, the authors provide evidence that the D-AVA pathway suppresses hyperosmolarity avoidance and enhances attractant chemotaxis. Finally, the authors present data implicating electrical coupling between the AVAL and AVAR neurons through the innexin UNC-9 as

essential for this motor feedback pathway and suggest the AVA neurons may therefore act as a hub for sensory integration during animal decision-making. Overall, the electrophysiology analysis is technically impressive and thorough, and the authors' findings are intriguing. However, the manuscript also includes experiments that are not well integrated into the overall message and therefore there is limited room for discussion of the authors' findings in a broader context. For example, GABAergic neurons have primary outputs onto muscles, but how muscle inhibition would factor into the authors' model is not at all addressed. The below suggestions would strengthen the manuscript.

1. The electrophysiological analysis of LGC-46 and cholinergic currents in GABAergic motor neurons is convincing. However, LGC-46 has previously been reported to localize to presynaptic cholinergic terminals and act as an anion channel (Takayanagi-Kiya, S, et al, 2016). More explanation and discussion (and acknowledgement of this alternate model) therefore seems warranted.
2. While not essential for the authors' main conclusion, it may also be interesting to examine subcellular localization of LGC-46 in GABAergic neurons. Several recent studies (Philbrook et al 2018, He et al. 2019, Cuentas-Condori, 2019) not cited by the authors show that ACR-12-containing receptors localize to dendritic spines on GABAergic neurons and that evoked calcium transients in GABAergic neurons are reduced by *acr-12* deletion in recordings from intact animals. It would be interesting to know if LGC-46 receptors also localize to these structures or represent a distinct population. Also, given that the dissection for electrophysiology likely disrupts at least some of the commissural projections, it seems possible that the authors observe only a subset of synaptic events in their recordings. This may account for the lack of effects with deletions in other subunits with known expression in GABAergic neurons. The authors should cite these additional papers and acknowledge this potential limitation in their approach.
3. While the authors show data for evoked responses in AVA following stimulation of GABAergic neurons, it is somewhat surprising they didn't measure the rate of IPSCs in AVA from wt versus RNA-treated animals. Such measurements would support the authors contention that these connections are active under normal conditions.
4. Figure 4: Incorporation of data for *unc-17* mutants/RNAi or RNAi of *lgc-46* in GABAergic neurons into the authors' analysis of glycerol induced currents in VD5 would strengthen the authors conclusions that the currents arise indirectly through presynaptic activation of cholinergic neurons rather than direct connections. Inclusion of such data would also serve to better integrate the findings for *lgc-46* in Figure 1 with the rest of the paper.
5. The pressure evoked experiments are not well integrated into the rest of the paper and dilute the main message. It is not clear how stretch responses in D neurons would fit with the authors' model. Additional clarification seems warranted.
6. Based on the authors' model, GABAergic inhibition of AVA should regulate the length or percentage of backward runs during escape. The authors use this sort of metric in Figures 1 and 2, but shift to other behavioral measures in subsequent figures. It is not completely clear how backward movement is affected in these behavioral assays. It would therefore be interesting to know how knockdown of *unc-49* in AVA impacts backward versus forward runs upon glycerol or diacetyl exposure.
7. There is limited discussion of how the authors findings fit with existing models for generation of forward and backward movement. For example, reciprocal connectivity across classes of interneurons, evidence for tyraminerbic inhibition of AVB during escape, and the contribution of GABAergic connectivity with muscles are not discussed.
8. The wiring data suggests GABAergic synapses onto AVA neurons are solely from VD neurons.

The authors should acknowledge this and incorporate into their model.

9. There are a few grammatical and spelling errors throughout the text. For example, thread→threat (intro) and thread-reward→ threat-reward (intro)

Reviewer #3 (Remarks to the Author):

The study in Liu et al characterized the functional connectivity and signaling processing of a worm circuit that transmitted the inhibitory GABA signal from D-type motor neurons to the interneurons AVA and identified the function of this feedback circuit in locomotion and chemotaxis behavior. The findings will generate advances to the understanding of the motor systems in regulating information integration to generate coordinated behavior. Specifically, the authors first found that an acetylcholine signal was required for the sPSC of D-MN and an acetylcholine receptor Igc-46 functions in D-MN to regulate the cholinergic neurotransmission. By optogenetically stimulating A- and B-MN, they further showed that A and B-MN regulate D-MN. Intriguingly, they found that inhibiting D-MN altered the balance between forward and backward movements by increasing backward movements. They further showed that D-MN likely regulated AVA through a type A GABA receptor UNC-49. These results establish the functional circuit from A/B-MN to D-MN to AVA. Based on these findings, they characterized the responses of the D -> AVA circuit to glycerol sensed by ASH and to diacetyl sensed by AWA, and how the two sensory signals integrate in AVA to produce a coordinated behavioral response. In addition, they showed that an ENaC channel unc-8 was responsible for the response of AVA to air puffs. Furthermore, the left and right AVA neurons are different in biophysical properties and the innexin unc-9 regulates the coupling of the two AVAs. Overall, the study aims to address an important question in the field of motor controls. The experiments in general are well-designed and well-controlled. It is compelling that the authors were able to identify the effects of manipulating the neurotransmission of the circuit in chemotaxis behaviors. The overall results are convincing. However, a few issues of the study need to be addressed.

Major:

The experimental approach that combines optogenetics and electrophysiological recording are powerful. They are nice examples of how the worm nervous system can be best used to study circuit and cell properties. The methods show that the non-retinal controls were performed. However, it is not clear whether the control results are included in the paper. These controls are critical for drawing the conclusions and need to be shown together with the experimental results.

To characterize the function of D -> AVA, in Figure 4 the authors showed that AVA responded to glycerol and the response required unc-49. Meanwhile, Vd5 responded to glycerol. Does the response in D depend on AVA or A-MN? Does Igc-46 RNAi in D abolish the response in Vd5? Or, does unc-17 RNAi in A abolish the response in D? Does the inhibition on AVA require the GABA signal from D? Do these genetic manipulations generate predicted behavioral defects? Similarly, in Figure 5 the authors showed that AVA responded to diacetyl and the response required unc-49. Vd5 also responded to diacetyl. However, it is not clear whether the response in D depends on B-MN. Addressing some of these questions will strengthen the functional characterization of A/B-MN -> D-MN -> AVA and its role in regulating chemotaxis behavior.

In Figure 6, the authors used glycerol and diacetyl together to investigate how these cues regulate AVA. It is a nice set of experiments that demonstrate how complex sensory inputs are integrated in AVA to regulate behavior. However, it does not strongly support the involvement and the function of motor integration in this process, because both of the AVA responses are generated by sensory cues. D-MN are likely to be important in motor integration as the authors nicely showed

above and the role of D in providing motor information in the integration process needs to be shown. Or, the authors need to better interpret the results of these experiments. Supplementary Figure 8 should be a main figure to help addressing the questions.

The authors showed that the ENaC channel *unc-8* regulated the response of AVA to air puffs and suggested that D-MNs used *UNC-8* to respond to stretch. There are a few concerns about this experiment and the conclusion. First, it is not clear why air puffs are similar to stretches. It is also not clear how the idea that D-MN respond to stretch relates to the rest of the paper. The authors need to clarify the importance of these findings and incorporate these experiments into the study.

The last part of the paper carefully characterized the difference between the left and the right AVA in their biophysical properties based on their difference in connectivities. The experiments are well done and the results are convincing. The findings are important. However, again, the authors need to better incorporate these results with the rest of the study.

Minor:

Some sentences are confusing and need to be clarified. For example: "A direct inhibition of VA5 by D-MNs would be manifested as an optogenetically evoked inward current whereas an inhibition of AVA would cause reduced frequency of spontaneous PSCs under our experimental conditions (holding voltage -60 mV, chloride equilibrium potential -6 mV). We found that blue light stimuli did not cause any appreciable inward current but inhibited spontaneous PSCs in VA5 profoundly (>90%) and reversibly (Fig. 2a)." and "The lack of a difference between wild type and the RNAi strain at 4 M glycerol was probably because this high concentration of glycerol greatly inhibited worm locomotion."

The defect of the *unc-49* mutants should be rescued by expressing *unc-49* in AVA.

Figure 2 is too small.

We thank the reviewers for the careful review of our manuscript, and for the thoughtful and constructive comments. We have carefully addressed them in the revised manuscript. Below are our point-by-point responses to the reviewers' comments. The line numbers indicated in our responses match those in the Merged PDF, which was generated by the manuscript submission system.

Reviewer #1

Comment

The authors aim to show that D-MNs are able to bias the worm's behavior toward reward-decision making by preventing avoidance in threat response. Specifically, they show that D-MNs inhibit upstream AVA via a GABA receptor, UNC-49. AVAL and R differ electrically and balance this inhibition to arrive at a decision. D-MNs receive two excitatory inputs: cholinergic MNs & current through a stretch receptor. They also identify receptors on D-MNs which mediate cholinergic MN synapses, and identify receptors on AVA mediating D-MN inhibition of AVA and the innexin responsible for coupling AVA-L & AVA-R. I do have a few concerns.

Response

We thank the reviewer for this general evaluation about our work.

Comment

1. A previous study Petrash et al J neurosci 33, 5524-5532 (2013) identify *acr-12* as a post-synaptic receptor for D-MNs for cholinergic MN synapses. How do the authors reconcile their results with this data?

Response

In this earlier study, ACR-12 was identified as a postsynaptic receptor in D-MNs based on indirect evidence, such as increased sensitivities of *acr-12* mutants to a paralytic effect of aldicarb (a cholinesterase inhibitor), reduced frequencies of spontaneous postsynaptic currents (sPSCs) recorded from body-wall muscle cells in an *acr-12* mutant, an abnormal locomotion waveform displayed by an *acr-12* mutant, and partial colocalization between a synaptic vesicle marker expressed in cholinergic MNs and GFP-tagged ACR-12 expressed in D-MNs. In our study, the measurements of spontaneous and optogenetically evoked PSCs in D-MNs allowed direct assessment of synaptic transmission from cholinergic MNs to D-MNs. In the revised manuscript, we discussed possible causes for the different conclusions between the two studies (lines 457-472).

Comment

2. The authors use optogenetic stimulation of GABAergic neurons and suggest that D-MNs are responsible? Aren't there additional GABA synapses on AVA?

Response

We used the *unc-47* (vesicular GABA transporter) promoter for GABAergic neuron-targeted expression of ChR2. Based on GFP reporter expression, this promoter has activities in only 26 neurons, including RME (4), AVL (1), RIS (1), DVB (1), D-MNs (19) (McIntire et al., Nature 389: 870-876, 1997). Among these neurons, only D-MNs (VD5, VD11, and VD13) provide chemical

synaptic inputs to AVA (Varshney et al., *PLoS Comput Biol* e1001066, 2017). We addressed this comment in the revised manuscript (lines 106-109, and lines 147-149).

Comment

3. Is the D-MN effect on AVA present in all instances of forward vs backward locomotion or is it specific to aversion/reward stimuli?

Response

The D-AVA circuit was tonically active, and modulated worm locomotion in either the absence or presence of an attractant and/or a repellent under our experimental conditions. Therefore, it likely plays a generally important role in modulating forward vs backward locomotion. To address the reviewer's comment, we added similar sentences to the Discussion (lines 407-415).

Comment

4. The authors stimulate ASH, which also has uneven inputs to AVAL and R, but do not observe the same differences on either side. These results should be discussed.

Response

We thank the reviewer for pointing this out. We made a mistake in describing the number of chemical synapses from ASH to AVA. Actually, the difference is small (10 to AVAR and 12 to AVAL). Furthermore, gap junctions exist between ASHL and ASHR. Therefore, it is not surprising that the strength of synaptic inputs from ASH appeared to be indistinguishable between AVAL and AVAR. We clarified about this in the revised manuscript (lines 347-356).

Comment

5. The authors the effect of gabazine on spontaneous activity in VA5 (downstream of D-MNs through AVA) and found that sPSCs were increased in a dosaged dependent way in VA5 so this is an active circuit at rest (disinhibited AVA, leading to excitation of VA5). This could also be due to other GABAergic inputs on VA5.

Response

Our previous study (Liu et al., *Nat Commun* 2017) showed that sPSCs in VA5 include two types: large and slow events caused by ACh release from AVA, and small and fast events caused by the activation of an ACh autoreceptor. In this study, we only quantified the large and slow events. Therefore, the quantified sPSCs only reflected synaptic transmission from AVA to VA5. This information was described in a figure legend in the original manuscript but is also described in the main text of the revised manuscript (lines 135-137, and lines 141-143).

Comment

6. The authors use a ring avoidance assay to prove the role of D-MN-AVA synapses. In this assay, we expect that when D-AVA circuit is disrupted in *unc-49 RNAi*, that we see an increase in backward locomotion. Forward locomotion we assume is necessary for escape. So, if D-AVA promotes forward locomotion then we should have more escaping worms. This appears to be true at intermediate glycerol concentrations, but not high?

Response

A significant effect of disrupting the D-AVA circuit on the glycerol escape response was observed at glycerol concentrations of 2 M and 3 M but not 4 M in the glycerol alone behavioral assay (Fig. 4e) but at all the glycerol concentrations (2-4 M) in the glycerol plus diacetyl behavioral assay (Fig. 6c). We speculate that the apparent lack of an effect of the D-AVA circuit at 4 M in the glycerol alone assay was probably because the motivation of worms to escape was outweighed by an inhibitory effect of glycerol on locomotion. We added a similar interpretation to the manuscript (lines 244-246).

Minor Comments

1. Supplemental Figure 4 is mis-written as Supplemental Figure 3 in the text.

Response: It is corrected in the revised manuscript.

2. Figure 6 legend mentions “Depolarization last time of AVA”, which is unclear.

Response: We agree with the reviewer that the phrase “last time” is confusing. It has been changed to “duration” in the revised manuscript.

3. Reference for AVAL has 1 D-MN synapse, and AVAR has 5 D-MN synapses.

Response: The reference (Varshney, et al., 2011) has been added to the revised manuscript (lines 309-311).

Reviewer #2

Comment

The integration of sensory inputs to make appropriate behavioral responses is well-known to be essential for organismal survival, but evidence that motor systems actively participate in shaping sensory responsiveness has only recently begun to emerge. Here Liu et al. use electrophysiology and genetic approaches to address questions about the circuit and molecular mechanisms that regulate responses to aversive and attractive stimuli in the nematode *C. elegans*. The authors provide evidence that D-class GABAergic motor neurons inhibit the backward locomotory circuit through inhibitory synaptic contacts onto AVA locomotory control interneurons that have primary roles in promoting backward movement. The authors use AVA-specific RNAi to knockdown expression of the GABAA receptor UNC-49 to show to argue that synaptic inhibition of AVA from GABAergic motor neurons biases animals toward forward locomotion during threat-reward decision making. In particular, the authors provide evidence that the D-AVA pathway suppresses hyperosmolarity avoidance and enhances attractant chemotaxis. Finally, the authors present data implicating electrical coupling between the AVAL and AVAR neurons through the innexin UNC-9 as essential for this motor feedback pathway and suggest the AVA neurons may therefore act as a hub for sensory integration during animal decision-making. Overall, the electrophysiology analysis is technically impressive and thorough, and the authors’ findings are intriguing. However, the manuscript also includes experiments

that are not well integrated into the overall message and therefore there is limited room for discussion of the authors' findings in a broader context. For example, GABAergic neurons have primary outputs onto muscles, but how muscle inhibition would factor into the authors' model is not at all addressed. The below suggestions would strengthen the manuscript.

Response

We thank the reviewer for this general evaluation about our work.

Comment

1. The electrophysiological analysis of LGC-46 and cholinergic currents in GABAergic motor neurons is convincing. However, LGC-46 has previously been reported to localize to presynaptic cholinergic terminals and act as an anion channel (Takayanagi-Kiya, S, et al, 2016). More explanation and discussion (and acknowledgement of this alternate model) therefore seems warranted.

Response

We thank the reviewer for this suggestion. Besides the presynaptic anion channel and the postsynaptic ACh receptor described by the previous and current studies, the results of our earlier study indicate that LGC-46 is also a key component of an ACh receptor in cholinergic A-MNs (Liu et al., *Nat Commun* 2017). However, LGC-46-dependent sPSCs have a much longer decay time in A-MNs than D-MNs (>30-fold difference), which is described in the revised manuscript (lines 120-122). The LGC-46 receptors presumably serves as cation channels in both A-MNs and D-MNs because they mediate excitatory synaptic transmission. The apparently different receptor functions reported by the three studies (cation versus anion selectivity, and slow versus fast kinetics) suggest that LGC-46 may coassemble with different proteins to form functionally distinct heteromeric ionotropic receptors. We discussed about this in the revised manuscript (lines 473-484).

Comment

2. While not essential for the authors' main conclusion, it may also be interesting to examine subcellular localization of LGC-46 in GABAergic neurons. Several recent studies (Philbrook et al 2018, He et al. 2019, Cuentas-Condori, 2019) not cited by the authors show that ACR-12-containing receptors localize to dendritic spines on GABAergic neurons and that evoked calcium transients in GABAergic neurons are reduced by *acr-12* deletion in recordings from intact animals. It would be interesting to know if LGC-46 receptors also localize to these structures or represent a distinct population. Also, given that the dissection for electrophysiology likely disrupts at least some of the commissural projections, it seems possible that the authors observe only a subset of synaptic events in their recordings. This may account for the lack of effects with deletions in other subunits with known expression in GABAergic neurons. The authors should cite these additional papers and acknowledge this potential limitation in their approach.

Response

To address these thoughtful comments of the reviewer, we did the followings: 1) we cited the three papers in the revised manuscript (line 464); 2) we analyzed the subcellular localization of GFP-tagged LGC-46 in D-MNs, and found that it partially colocalized with a presynaptic

marker expressed in cholinergic MNs (Fig. 1e), which is consistent with a role of LGC-46 in the synaptic transmission from cholinergic MNs to D-MNs; and 3) we speculated that our failure to detect an effect of *acr-12* deficiency on sPSCs in D-MNs might be due to the disruption of some commissural projections (lines 368-370).

Comment

3. While the authors show data for evoked responses in AVA following stimulation of GABAergic neurons, it is somewhat surprising they didn't measure the rate of IPSCs in AVA from wt versus RNA-treated animals. Such measurements would support the authors contention that these connections are active under normal conditions.

Response

As suggested by the reviewer, we recorded spontaneous IPSCs in AVA from wild type, *unc-49(e407)* mutant, and the AVA-targeted *unc-49* RNAi strain. Both the mutant and the RNAi strain showed much lower IPSC frequencies compared with wild type (Fig. 2d). The results further substantiate our conclusion that UNC-49 is the GABA receptor in AVA.

Comment

4. Figure 4: Incorporation of data for *unc-17* mutants/RNAi or RNAi of *lgc-46* in GABAergic neurons into the authors' analysis of glycerol induced currents in VD5 would strengthen the authors conclusions that the currents arise indirectly through presynaptic activation of cholinergic neurons rather than direct connections. Inclusion of such data would also serve to better integrate the findings for *lgc-46* in Figure 1 with the rest of the paper.

Response

As suggested by the reviewer, we added results from the GABAergic neuron-targeted *lgc-46* RNAi strain (Fig. 4c-e). These new results provide further support to our conclusion.

Comment

5. The pressure evoked experiments are not well integrated into the rest of the paper and dilute the main message. It is not clear how stretch responses in D neurons would fit with the authors' model. Additional clarification seems warranted.

Response

Our results indicate that the D-AVA circuit may be activated by either chemical synaptic transmission from cholinergic MNs or activation of the UNC-8 stretch receptor in D-MNs. Under physiological conditions, the UNC-8 stretch receptor in D-MNs is likely activated by worm body bending, like the putative stretch receptor in B-type cholinergic MNs (Wen et al., *Neuron* 76: 750-761, 2012). This property may help the D-AVA circuit regulate locomotion behaviors. We added similar comments to the revised manuscript (lines 452-456).

Comment

6. Based on the authors' model, GABAergic inhibition of AVA should regulate the length or percentage of backward runs during escape. The authors use this sort of metric in Figures 1 and 2, but shift to other behavioral measures in subsequent figures. It is not completely clear how backward movement is affected in these behavioral assays. It would therefore be interesting to

know how knockdown of *unc-49* in AVA impacts backward versus forward runs upon glycerol or diacetyl exposure.

Response

To address the reviewer's comment, we quantified locomotor kinematics of freely moving worms under the experimental conditions for assaying the decision-making behavior. We found that disruption of the D-AVA circuit by AVA-targeted *unc-49* RNAi caused reduced forward locomotion but increased backward locomotion (Fig. 7d), which supports our conclusion that a function of the D-AVA circuit is to promote the escape response by promoting forward movement when worms are confronted with both an attractant and a repellent.

Comment

7. There is limited discussion of how the authors findings fit with existing models for generation of forward and backward movement. For example, reciprocal connectivity across classes of interneurons, evidence for tyraminerpic inhibition of AVB during escape, and the contribution of GABAergic connectivity with muscles are not discussed.

Response

To address this comment of the reviewer, we did the followings: 1) we described reciprocal connectivity between AVA and AVB, direct or indirect synaptic connections between the tyraminerpic interneurons RIM and other neurons (e. g. AVA, AVB, and primary sensory neurons), and potential functional interactions between the A-AVA circuit and other neurons in the worm's locomotion circuit (lines 385-393); and 2) we discussed the contribution of GABAergic connectivity with muscles to locomotion (lines 394-406).

Comment

8. The wiring data suggests GABAergic synapses onto AVA neurons are solely from VD neurons. The authors should acknowledge this and incorporate into their model.

Response

In the revised manuscript, we clarified that several VDs (VD5, VD6, VD11 and VD13) are presynaptic to AVA in the worm's wiring diagram (lines 309-311). In addition, we changed "D" to "VD" in all the related circuit diagrams (Figs. 2g, 4a, 5a, 6a, and 7a).

Comment

9. There are a few grammatical and spelling errors throughout the text. For example, thread→threat (intro) and thread-reward→ threat-reward (intro).

Response

We have fixed them.

Reviewer #3

Comment

The study in Liu et al characterized the functional connectivity and signaling processing of a worm circuit that transmitted the inhibitory GABA signal from D-type motor neurons to the interneurons AVA and identified the function of this feedback circuit in locomotion and

chemotaxis behavior. The findings will generate advances to the understanding of the motor systems in regulating information integration to generate coordinated behavior. Specifically, the authors first found that an acetylcholine signal was required for the sPSC of D-MN and an acetylcholine receptor *Igc-46* functions in D-MN to regulate the cholinergic neurotransmission. By optogenetically stimulating A- and B-MN, they further showed that A and B-MN regulate D-MN. Intriguingly, they found that inhibiting D-MN altered the balance between forward and backward movements by increasing backward movements. They further showed that D-MN likely regulated AVA through a type A GABA receptor *UNC-49*. These results establish the functional circuit from A/B-MN to D-MN to AVA. Based on these findings, they characterized the responses of the D → AVA circuit to glycerol sensed by ASH and to diacetyl sensed by AWA, and how the two sensory signals integrate in AVA to produce a coordinated behavioral response. In addition, they showed that an ENaC channel *unc-8* was responsible for the response of AVA to air puffs. Furthermore, the left and right AVA neurons are different in biophysical properties and the innexin *unc-9* regulates the coupling of the two AVAs. Overall, the study aims to address an important question in the field of motor controls. The experiments in general are well-designed and well-controlled. It is compelling that the authors were able to identify the effects of manipulating the neurotransmission of the circuit in chemotaxis behaviors. The overall results are convincing. However, a few issues of the study need to be addressed.

Response

We thank the reviewer for this general evaluation about our work.

Major:

Comment

The experimental approach that combines optogenetics and electrophysiological recording are powerful. They are nice examples of how the worm nervous system can be best used to study circuit and cell properties. The methods show that the non-retinal controls were performed. However, it is not clear whether the control results are included in the paper. These controls are critical for drawing the conclusions and need to be shown together with the experimental results.

Response

Because none of the recordings from the non-retinal groups showed any effect, we did not include these results in the original figures for clarity. All the related figures have been revised to include these results.

Comment

To characterize the function of D → AVA, in Figure 4 the authors showed that AVA responded to glycerol and the response required *unc-49*. Meanwhile, *Vd5* responded to glycerol. Does the response in D depend on AVA or A-MN? Does *Igc-46* RNAi in D abolish the response in *Vd5*? Or, does *unc-17* RNAi in A abolish the response in D? Does the inhibition on AVA require the GABA signal from D? Do these genetic manipulations generate predicted behavioral defects? Similarly, in Figure 5 the authors showed that AVA responded to diacetyl and the response required *unc-49*. *Vd5* also responded to diacetyl. However, it is not clear whether the response in D depends

on B-MN. Addressing some of these questions will strengthen the functional characterization of A/B-MN -> D-MN -> AVA and its role in regulating chemotaxis behavior.

Response

We thank the reviewer for these thoughtful comments. To address the reviewer's comments about Fig. 4, we performed several additional experiments with the D-MN-targeted *lgc-46* RNAi strain. The new results show that 1) the activating effect of glycerol on VD5 was substantially less in the *lgc-46* RNAi strain than wild type (Fig. 4c); and 2) similar to the AVA-targeted *unc-49* RNAi strain, the *lgc-46* RNAi strain exhibited a larger degree of glycerol-evoked AVA membrane depolarization and a weaker glycerol escape response compared with wild type (Fig. 4d, e). These results further substantiate our conclusions about the function of the D-AVA circuit. For Fig. 5, the reviewer wonders whether the response in D-MNs depends on B-MNs. Based on existing knowledge, activation of AVA by diacetyl would cause an increased activity of the forward locomotion neural circuit (Ghosh et al., *Neuron* 92: 1049-1062, 2016; Shrinkai, et al., *J Neurosci* 31: 3007-3015, 2011). Because forward movement depends on increased activities of B-MNs (de Bono and Maricq. *Annu Rev Neurosci*, 28: 451-501, 2005), and the latter is the most important source of presynaptic inputs to D-MNs (White et al. *Philos Trans R Soc Lond B Biol Sci* 314: 1-340, 1986), the diacetyl-induced VD5 depolarization and PSC bursts are presumably due to synaptic transmission from B-MNs. We addressed the reviewer's comment by stating that the diacetyl-induced depolarization of and PSC bursts in VD5 "presumably resulted from excitatory synaptic transmission from B-MNs because they are the primary source of chemical synaptic inputs to D-MNs, although we cannot exclude potential contributions from other neurons." (lines 254-256).

Comment

In Figure 6, the authors used glycerol and diacetyl together to investigate how these cues regulate AVA. It is a nice set of experiments that demonstrate how complex sensory inputs are integrated in AVA to regulate behavior. However, it does not strongly support the involvement and the function of motor integration in this process, because both of the AVA responses are generated by sensory cues. D-MN are likely to be important in motor integration as the authors nicely showed above and the role of D in providing motor information in the integration process needs to be shown. Or, the authors need to better interpret the results of these experiments. Supplementary Figure 8 should be a main figure to help addressing the questions.

Response

We took the second approach suggested by the reviewer. Specifically, 1) we moved the original Supplementary Figure 8 into the main body of the manuscript (now Fig. 6); and 2) we described that "D-MNs might integrate excitatory synaptic signals from cholinergic MNs and UNC-8 stretch receptor-sensed body bending information to modulate locomotion behavior and threat-reward decision making through the D-AVA circuit" (lines 454-456).

Comment

The authors showed that the ENaC channel *unc-8* regulated the response of AVA to air puffs and suggested that D-MNs used UNC-8 to respond to stretch. There are a few concerns about this experiment and the conclusion. First, it is not clear why air puffs are similar to stretches. It is also not clear how the idea that D-MN respond to stretch relates to the rest of the paper. The

authors need to clarify the importance of these findings and incorporate these experiments into the study.

Response

We pressure-ejected the bath solution (not air) through a glass pipette aimed at the dendrite of VD5. Pressure ejection of the bath solution is a commonly used approach to activate stretch-sensitive receptors (*e. g.* Connelly et al., PNAS 112: 590-595, 2015; Sullivan et al., Circ Res 80: 861-867, 1997; Woo et al., *Cell Calcium* 41: 397-403, 2007). In the revised manuscript, we made it clearer that we pressure-ejected the bath solution to activate the stretch receptor. In addition, we discuss why activation of the stretch receptor in D-MNs are important to the function of the D-AVA circuit (449-456).

Comment

The last part of the paper carefully characterized the difference between the left and the right AVA in their biophysical properties based on their difference in connectivities. The experiments are well done and the results are convincing. The findings are important. However, again, the authors need to better incorporate these results with the rest of the study.

Response

In response to this comment, we considered more deeply about how the different biophysical properties of AVAL and AVAR might be related to the D-AVA circuit. As suggested by the larger membrane voltage changes observed in AVAL than AVAR in response to current injections (Fig. 8b), AVAL has a higher membrane resistance than AVAR. This difference is expected to partially compensate for the weaker synaptic inputs from D-MNs to AVAL than AVAR. Therefore, it is an additional mechanism, besides the electrical coupling, that helps the two AVA neurons produce similar membrane voltage changes in response to different strengths of synaptic inputs from D-MNs, and possibly from some other neurons as well. We added these interpretations to the revised manuscript (lines 424-428).

Minor:

Comment

Some sentences are confusing and need to be clarified. For example: “A direct inhibition of VA5 by D-MNs would be manifested as an optogenetically evoked inward current whereas an inhibition of AVA would cause reduced frequency of spontaneous PSCs under our experimental conditions (holding voltage -60 mV, chloride equilibrium potential -6 mV). We found that blue light stimuli did not cause any appreciable inward current but inhibited spontaneous PSCs in VA5 profoundly (>90%) and reversibly (Fig. 2a).” and “The lack of a difference between wild type and the RNAi strain at 4 M glycerol was probably because this high concentration of glycerol greatly inhibited worm locomotion.”

Response

We apologize for the confusion. We have edited the two sentences to make them clearer. They now read as “Under our experimental conditions, optogenetic activation of an inotropic GABA receptor in VA5 by stimulation of GABAergic neurons would manifest as an evoked inward current because the Cl⁻ equilibrium potential (-6 mV) was more depolarized than the holding potential (-60 mV), whereas optogenetic inhibition of AVA would cause a reduced frequency of

the slow and large sPSCs in VA5.” (lines 137-141), and “However, a significant difference was not detected at 4M glycerol, which was probably because the motivation of worms to escape was outweighed by an inhibitory effect of glycerol on locomotion.” (lines 244-246).

Comment

The defect of the *unc-49* mutants should be rescued by expressing *unc-49* in AVA.

Response

During the course of this study, we initially had the same thought to do this experiment but later decided against it. Since *unc-49* is expressed in many other cells, including body-wall muscle cells and some other neurons, results from an AVA-targeted rescue strain would be of limited value for interpreting physiological roles of the D-AVA circuit. On the other hand, the AVA-targeted *unc-49* RNAi strain allowed us not only to confirm the effect of the *unc-49* mutation on synaptic transmission but also to assess how a specific disruption of the D-AVA circuit may alter behavior.

Comment

Figure 2 is too small.

Response

We agree with the reviewer that it is desirable to make Fig. 2 bigger. After making a few minor changes to the figure, all the figure contents appear to be legible in the printout (two-column width).

REVIEWERS' COMMENTS

Reviewer #1 (Remarks to the Author):

The authors have done a reasonable job addressing my concerns. I do have a few suggestions that would help improve the manuscript.

1. I would recommend editing the section on the effect of gabazine on the spontaneous activity in VA5 neurons. I previously asked the authors to comment on the role of GABAergic inputs on VA5 neurons. The authors responded by saying that sPSCs are of two kinds (large and slow; small and fast), which they previously published (Lie et al 2017). They modified the text in response. However, I feel that these edits are a bit confusing with respect to their published work. Their prior work does not say that sPSCs can be subdivided. Instead, it says PSCs can be subdivided into sPSCs ("s" for slow) and fPSCs ("f" for fast). The sPSCs appear to have a decay time of about 10-100ms, with a current amplitude on average of about 12 pA, while the fPSCs have short decays <5ms or so, with amplitudes on average of about 4 pA. They show in their prior work that AVA affects VA5's sPSCs.

I would change the text to say: We previously showed that PSCs in VA5 may be divided into two types: slow and large events (sPSCs) caused by ACh release from AVA, and fast and small events (fPSCs) caused by the activation of an ACh autoreceptor(34).

2. The Methods "Data Analysis" section states is that sPSCs were quantified and there is no distinction on what the heuristics are specifically. I would recommend that the authors include a description of the parameters used. Also, it would nice to know how sPSC and fPSC are distinguished.

3. The authors admit that D-MN influence on AVA is a general property of modulating forward vs. backward movement. Furthermore, they add "Our results showed the D-AVA circuit was tonically active, and modulated worm locomotion in either the absence or presence of an attractant and/or a repellent. Conceivably, the D-AVA circuit may also have tonic activities under natural conditions and play a generally important role in modulating forward vs backward locomotion". If this is the case then it might be more appropriate for the title to be "GABAergic motor neurons bias locomotor decision making in *C. elegans*" if the phenomenon is a general property of locomotion and not about threat and reward.

4. I strongly urge the authors to edit their manuscript for clarity, particularly for a non-*C. elegans* reader.

Reviewer #2 (Remarks to the Author):

Overall, the authors have done a thorough job of addressing my prior review and I support publication.

I have only one minor comment. It is surprising that so much of the LGC-46::GFP fluorescent signal in Figure 1E is not apposed apposed by presynaptic ELKS-1. Might this arise as a consequence of overexpression? The authors should comment.

Reviewer #3 (Remarks to the Author):

The revised manuscript fully addressed all of my concerns.

We appreciate the careful review of our revised manuscript by the reviewers. Below are our point-by-point responses to their comments.

Reviewer #1

1. **Comment:** I would recommend editing the section on the effect of gabazine on the spontaneous activity in VA5 neurons. I previously asked the authors to comment on the role of GABAergic inputs on VA5 neurons. The authors responded by saying that sPSCs are of two kinds (large and slow; small and fast), which they previously published (Lie et al 2017). They modified the text in response. However, I feel that these edits are a bit confusing with respect to their published work.

Their prior work does not say that sPSCs can be subdivided. Instead, it says PSCs can be subdivided into sPSCs ("s" for slow) and fPSCs ("f" for fast). The sPSCs appear to have a decay time of about 10-100ms, with a current amplitude on average of about 12 pA, while the fPSCs have short decays <5ms or so, with amplitudes on average of about 4 pA. They show in their prior work that AVA affects VA5's sPSCs.

I would change the text to say: We previously showed that PSCs in VA5 may be divided into two types: slow and large events (sPSCs) caused by ACh release from AVA, and fast and small events (fPSCs) caused by the activation of an ACh autoreceptor(34).

Response: We have modified our original sentence to exactly what the reviewer suggested. In the original manuscript, the term "sPSCs" was also used in a few other places to stand for "spontaneous PSCs". To avoid a confusion between the slow and large events in VA5 and the spontaneous PSCs recorded from other neurons, "sPSCs" has been changed to "spontaneous PSCs" where it does not refer to events recorded from VA5.

2. **Comment:** The Methods "Data Analysis" section states is that sPSCs were quantified and there is no distinction on what the heuristics are specifically. I would recommend that the authors include a description of the parameters used. Also, it would nice to know how sPSC and fPSC are distinguished.

Response: As suggested, we added a description about the criterion used to identify the sPSCs in the Data Analysis part of the Methods section. sPSCs and fPSCs were distinguished based on whether their decay times were smaller or larger than 5 ms.

3. **Comment:** The authors admit that D-MN influence on AVA is a general property of modulating forward vs. backward movement. Furthermore, they add "Our results showed the D-AVA circuit was tonically active, and modulated worm locomotion in either the absence or presence of an attractant and/or a repellent. Conceivably, the D-AVA circuit may also have tonic activities under natural conditions and play a generally important role in modulating forward vs backward locomotion". If this is the case then it might be more appropriate for the title to be "GABAergic motor neurons bias locomotor decision making in *C. elegans*" if the phenomenon is a general property of locomotion and not about threat and reward.

Response: We thank the reviewer for this thoughtful comment. The title has been revised as suggested.

4. Comment: I strongly urge the authors to edit their manuscript for clarity, particularly for a non-C. elegans reader.

Response: We considered about this comment of the reviewer carefully. The names of C. elegans neurons and their wiring relationships may seem unfamiliar to some non-C. elegans readers. However, it is difficult to describe our findings accurately without using them.

Reviewer #2

Comment: I have only one minor comment. It is surprising that so much of the LGC-46::GFP fluorescent signal in Figure 1E is not apposed by presynaptic ELKS-1. Might this arise as a consequence of overexpression? The authors should comment.

Response: We thank the reviewer for this comment. We agree with the reviewer that some of the LGC-46::GFP puncta not colocalized with or juxtaposed to the TagRFP::ELKS-1 puncta could be due to overexpression of LGC-46::GFP. We added this interpretation to the Results section.

Reviewer #3

Comment: The revised manuscript fully addressed all of my concerns.